# Comprehensive molecular characterization of pediatric radiation-induced high-grade glioma

John DeSisto[1,13], John T. Lucas Jr.[2,13], Ke Xu[3], Andrew Donson[1], Tong Lin[4], Bridget Sanford[1], Gang Wu[3], Quynh T. Tran[5], Dale Hedges[2], Chih-Yang Hsu[2], Gregory T. Armstrong[6,7], Michael Arnold[8], Smita Bhatia[7,9], Patrick Flannery[1], Rakeb Lemma[1], Lakotah Hardie[10], Ulrich Schüller[11], Sujatha Venkataraman[1], Lindsey M. Hoffman[1,10], Kathleen Dorris[1,10], Jean M. Mulcahy Levy[1,10], Todd C. Hankinson[1,10], Michael Handler[1,10], Arthur K. Liu[10], Nicholas Foreman[1,10], Rajeev Vibhakar[1,10], Kenneth Jones[1], Sariah Allen[5], Jinghui Zhang[3], Suzanne J. Baker[12], Thomas E. Merchant[2], Brent A. Orr[5,14✉] & Adam L. Green[1,10,14✉]

Radiation-induced high-grade gliomas (RIGs) are an incurable late complication of cranial radiation therapy. We performed DNA methylation profiling, RNA-seq, and DNA sequencing on 32 RIG tumors and an in vitro drug screen in two RIG cell lines. We report that based on DNA methylation, RIGs cluster primarily with the pediatric receptor tyrosine kinase I high-grade glioma subtype. Common copy-number alterations include Chromosome (Ch.) 1p loss/1q gain, and Ch. 13q and Ch. 14q loss; focal alterations include *PDGFRA* and *CDK4* gain and *CDKN2A* and *BCOR* loss. Transcriptomically, RIGs comprise a stem-like subgroup with lesser mutation burden and Ch. 1p loss and a pro-inflammatory subgroup with greater mutation burden and depleted DNA repair gene expression. Chromothripsis in several RIG samples is associated with extrachromosomal circular DNA-mediated amplification of *PDGFRA* and *CDK4*. Drug screening suggests microtubule inhibitors/stabilizers, DNA-damaging agents, MEK inhibition, and, in the inflammatory subgroup, proteasome inhibitors, as potentially effective therapies.

[1] Morgan Adams Foundation Pediatric Brain Tumor Research Program, University of Colorado School of Medicine, Aurora, CO, USA. [2] Department of Radiation Oncology, St. Jude Children's Research Hospital, Memphis, TN, USA. [3] Department of Computational Biology, St. Jude Children's Research Hospital, Memphis, TN, USA. [4] Department of Biostatistics, St. Jude Children's Research Hospital, Memphis, TN, USA. [5] Department of Pathology, St. Jude Children's Research Hospital, Memphis, TN, USA. [6] Department of Epidemiology and Cancer Control, St. Jude Children's Research Hospital, Memphis, TN, USA. [7] Childhood Cancer Survivor Study, St. Jude Children's Research Hospital, Memphis, TN, USA. [8] Department of Pathology, Nationwide Children's Hospital, Columbus, OH, USA. [9] Division of Hematology and Oncology, University of Alabama at Birmingham, Birmingham, AL, USA. [10] Children's Hospital Colorado, Aurora, CO, USA. [11] Institute of Neuropathology and Department of Pediatric Hematology and Oncology, University Medical Center, Children's Cancer Center, Hamburg, Germany. [12] Department of Developmental Neurobiology, St. Jude Children's Research Hospital, Memphis, TN, USA. [13] These authors contributed equally: John DeSisto, John T. Lucas Jr. [14] These authors jointly supervised this work: Brent A. Orr, Adam L. Green. ✉email: brent.orr@stjude.org; adam.green@cuanschutz.edu

Therapeutic radiation of the central nervous system to treat childhood cancers causes secondary damage to normal tissue, including non-lethal DNA double-stranded breaks (DSBs) that may trigger erroneous or incomplete DNA repair, particularly when the repair occurs via error-prone non-homologous end joining. Chemotherapies (e.g., cisplatin or doxorubicin) that produce DNA DSBs or replication inhibitors (e.g., gemcitabine or topoisomerase II inhibitors) may also cause errors in DNA repair[1,2]. Alterations in genomic DNA after erroneous DNA repair may produce pathogenic mutations contributing to the formation of a new radiation-induced malignancy[3].

Tumors known as radiation-induced glioblastomas or radiation-induced gliomas arise in ~3% of pediatric cancer survivors after cranial radiotherapy[4]. Given that all these tumors are high-grade gliomas but that their pathology includes a wider range of high-grade gliomas than glioblastoma alone, we use the term "radiation-induced high-grade glioma" in this study but have continued use of the RIG abbreviation. RIG is a rare but significant cause of late mortality in childhood cancer survivors[5], with limited therapeutic options[6,7]. Prior genomic analyses in small RIG cohorts identified greater copy-number alterations than in de novo pediatric high-grade glioma (pHGG), including focal amplification of PDGFRA and Ch. 1q gain[8,9]. These studies also noted mutations in TP53 and genes involved in receptor tyrosine and MAP kinase pathways. A study of gene expression in RIG suggested that overexpression of ERBB3 and SOX10 is more frequent in RIG than pHGG[10]. Treatment-related second cancers outside the brain similarly show recurrent genomic copy-number variations[11–14].

In this multi-institutional study, we evaluate the largest RIG cohort to date through comprehensive clinical and molecular analyses. For each RIG case, we identify the initial malignancy and its corresponding treatment along with the resultant RIG location, type, histology, and treatment history. We conduct comprehensive molecular profiling of RIGs using whole-genome or whole-exome sequencing (WES/WGS), 450 K/850 K methylation arrays, and RNA-seq. WES/WGS analyses are conducted on matched blood samples when available. We identify previously unknown molecular characteristics and propose a method for grouping RIG tumors by using a combination of genomic alterations, methylation profiling, and gene expression. Our analyses differentiate RIG from de novo pHGG and provide a rationale for developing alternative therapeutic approaches that we explore using in silico and in vitro preclinical therapeutic screening studies.

## Results

**Patient and disease characteristics.** The median age at diagnosis of first cancer in the RIG cohort was 7 years (range, 0.16–19). The most common initial diagnoses included medulloblastoma ($n = 12$; 38%), acute lymphoblastic leukemia (ALL) ($n = 10$; 31%), astrocytoma ($n = 3$; 9%), and ependymoma ($n = 2$; 6%) (Table 1 and Supplementary Table 1). Complete clinicopathologic data on treatment history, including radiotherapy directed at the initial tumor, were assembled for 87.5% of cases (Fig. 1a, Table 1, Supplementary Fig. 1, and Supplementary Tables 1 and 2). Two patients had known mismatch repair deficiencies resulting from a heterozygous germline loss of MSH2 (Case 2) and PMS2 (Case 32), respectively, but developed new tumors with sufficient latency and relationship to the radiotherapy field to be potentially considered radiation attributable. The median time between initial treatment and RIG diagnosis was 8.0 years (95% CI 7.3–13.2 years) (Fig. 1b). At the time of this analysis, all RIG patients with known outcomes were deceased ($n = 26$); six had

unknown clinical status (Fig. 1c). Median survival in the RIG cohort was 9 months (95% CI 5–24.7 months) (Fig. 1c and Supplementary Table 2), which is consistent with outcomes from prior published clinical series of RIG (Supplementary Table 2)[7,11]. RIG tumor grades were WHO grade III ($n = 3$, 9%), grade IV ($n = 18$, 56%), and pHGG not otherwise specified ($n = 11$, 34%) (Supplementary Table 1). Location of the RIG relative to the initial radiotherapy field was in-field but outside the high-dose region in 32% ($n = 8$) of evaluable cases and in the high-dose (prescribed dose) region in 68% of cases ($n = 17$) (Supplementary Data 1). The RIG cases in our cohort involved the frontal and occipital lobes and posterior fossa (cerebellum and inferior brainstem) regions (Fig. 1d and Supplementary Fig. 2A, B). All cases had cranial exposure to ionizing radiation (see "Methods" and Supplementary Fig. 1).

**RIGs primarily cluster with the pedRTK I methylation group.** DNA methylation profiling was performed and analyzed by hierarchical unsupervised clustering and t-distributed stochastic neighbor embedding (t-SNE) of 34 RIG samples from 31 tumors against a combined reference cohort of pediatric and adult CNS malignancies[15,16]. Of 31 cases, 25 RIG cases clustered among a group of epigenetically similar tumors consisting of H3K27M-negative midline pHGG and pediatric receptor tyrosine kinase I (pedRTK I) subgroup pHGG (Fig. 2a–c and Supplementary Fig. 3A–C)[17]. Methylation profiling could not distinguish RIGs from IDHwt, GBM-MID, and pedRTK I pHGG samples included in the reference cohorts[15]. Accordingly, we designate the predominant group of RIG cases as "pedRTK I" hereafter (Supplementary Table 3). The six RIGs that clustered with other methylation subgroups are detailed in Supplementary Data 1 and 2. The MGMT promoter was methylated in eight of 31 (25.8%) cases. There was no trend associating initial cancer diagnosis, latency, or survival (Supplementary Figs. 4 and 5B) with RIG methylation class (Supplementary Fig. 4A–C)[18]. The odds of being assigned to the pedRTK I subgroup in the RIG cohort were 29.2 (95% CI 9.15–92.9) times those of being assigned to that subgroup in the de novo pHGG cohort ($P < 0.0001$) (Supplementary Fig. 5A and Supplementary Data 3). Thus, RIGs predominantly align within the pedRTK I methylation subgroup despite disparate clinical origins.

**RIGs show recurrent copy-number abnormalities.** Given the classification of most of the RIG cohort into one pHGG methylation subgroup, gene and copy-number alterations and transcriptomic profiles of RIG samples relative to pHGG were analyzed to better understand their similarities and differences. RIG methylation data were reviewed for large segment (>25% of chromosome arm) and focal (≤3 Mb) copy-number alterations. Recurrent large segment alterations in RIG pedRTK I cases included Ch.1p loss (10/25, 40.0%), Ch.1q gain (13/25, 50%), Ch.13q loss (10/25, 40.0%), and Ch.14q loss (10/25, 40.0%) (Fig. 3a, Supplementary Data 1, 20, and Supplementary Figs. 6 and 7). PDGFRA gain/amplification (11/31, 35.5%), CDK4 amplification (6/31, 19.4%), CDKN2A loss (9/31, 29%), and BCOR loss (7/31, 22.6%) were common focal alterations (Supplementary Fig. 7, and Supplementary Data 6 and 18). For cases where methylation and WGS data were both available, copy-number change estimates were comparable between the two methods (Supplementary Data 4, 5, 18, and 19).

Focal copy-number variations (CNVs) in the RIG cohort were compared to those in de novo pHGG in patients enrolled in the HERBY phase II open-label, randomized, multicenter clinical trial of bevacizumab[18]. Total CNVs were significantly increased in RIG pedRTK I cases relative to de novo pHGG cases (Supplementary

**Table 1 Summary of RIG samples included in the study.**

| ID | First cancer | | Rad dose/field | | | | | | RNA | DNA | |
|---|---|---|---|---|---|---|---|---|---|---|---|
| | Age | His | Dose (Gy) | Field | Lat (y) | Status | Clin | RIG | Seq | Seq | Meth |
| 1 | – | ALL | – | – | – | – | No | HGG | – | WES | EPIC |
| 2 | 14.3 | G,O | 59.4 | Focal | 9 | D | Yes | GBM | – | WES | EPIC |
| 2B | – | G,O | – | – | – | – | Yes | – | – | – | EPIC |
| 5 | 3.6 | C | 54 | Focal | 12 | D | No | GBM | – | WES | EPIC |
| 6 | 7.1 | M | – | – | 6 | D | No | HGG | – | – | EPIC |
| 7 | – | M | – | – | – | – | No | HGG | – | WES | – |
| 8 | – | Ge | – | – | – | – | No | HGG | – | WES | EPIC |
| 9 | – | M | – | – | – | – | Yes | HGG | – | WES | EPIC |
| 11 | 0.16 | G,O | – | Focal | 13 | D | Yes | HGG | – | – | EPIC |
| 12 | 3.5 | ALL | 24 | CR | 13 | D | Yes | GBM | – | WGS | EPIC |
| 12B | – | ALL | – | – | – | – | Yes | – | – | – | 450 K |
| 13 | 0.3 | G,O | 49.5 | Focal | 3 | D | Yes | HGG | – | – | EPIC |
| 14 | 1.5 | ALL | 18 | CSI | 8 | D | Yes | GBM | – | – | EPIC |
| 14B | – | ALL | – | – | – | – | Yes | – | – | – | EPIC |
| 15 | 12.4 | Ge | 51 | Focal | 7 | D | Yes | HGG | – | – | EPIC |
| 16 | 5.8 | M | X, 55.8 | CSI | 7 | D | Yes | GBM | – | – | EPIC |
| 18 | 10.2 | M | 23.4, 55.8 | Focal | 8 | D | Yes | AA | – | – | EPIC |
| 19 | 8 | M | X, 55.8 | Focal | 3 | D | Yes | HGG | – | – | EPIC |
| 20 | 15 | Ga | – | Focal | 4 | – | Yes | GBM | – | – | EPIC |
| 21 | 7 | BL | – | CSI | 7 | D | Yes | GBM | A | WGS | EPIC |
| 22 | – | ALL | – | TBI | – | D | Yes | GBM | B | – | EPIC |
| 23 | 8 | M | 24, 54–55.8 | CSI | 4 | D | Yes | GBM | B | WGS | EPIC |
| 24 | 10 | M | 23.4, 54 | CSI | 13 | D | Yes | GBM | B | WGS | EPIC |
| 25 | 11 | E | 54 | Focal | 12 | D | Yes | GBM | A | – | EPIC |
| 26 | 3 | ALL | – | CR | 13 | D | Yes | GBM | A | – | EPIC |
| 27 | 4 | M | 23.4, 54 | CSI | 7 | D | Yes | HGG | A | WGS | EPIC |
| 28 | 19 | M | 36, 54 | CSI | 7 | D | Yes | GBM | B | WGS | EPIC |
| 29 | 3 | ALL | 21 | CR | 4 | D | Yes | GBM | B | WGS | EPIC |
| 30 | 2 | ALL | 18 | CR | 10 | D | Yes | GBM | A | WGS | EPIC |
| 31 | 4 | M | X, 54–55.8 | CSI | 7 | D | Yes | GBM | B | WGS | EPIC |
| 32 | 9 | ALL | 12 | TBI | – | D | Yes | AA | A | WGS | EPIC |
| 33 | 7 | M | 23.4, 54 | CSI | 10 | D | No | GBM | B | WGS | EPIC |
| 34 | 6 | ALL | – | TBI | 8 | D | Yes | AA | – | WGS | 450 K |
| 42 | 8.8 | E | – | – | 23 | – | No | HGG | – | – | EPIC |
| 43 | 2.3 | ALL | – | – | 11 | D | No | HGG | – | – | EPIC |

Histology (His): *ALL* acute lymphoblastic leukemia, *BL* Burkitt lymphoma, *C* craniopharyngioma, *E* ependymoma, *Ga* ganglioglioma, *Ge* germinoma, *G,O* glial, other, *M* medulloblastoma.
Clin: Clinical history reviewed (Yes/No); RIG: *AA* anaplastic astrocytoma, *GBM* glioblastoma, *HGG* high-grade glioma.
RNA-Seq: *A* Subgroup A tumor with RNA-Seq data, *B* Subgroup B tumor with RNA-Seq data.

Fig. 5C). *PDGFRA* amplification ($P = 0.188$) and *BCOR* deletions ($P = 0.0043$) were also enriched in RIG relative to de novo pHGG (Supplementary Data 6, 18, 19 and Supplementary Fig. 7). Changes in copy number of *PDGFRA* and *BCOR* were associated with an accompanying change in gene expression in cases for which RNA-seq data were available ($n = 12$, 34.3%) (Fig. 3b). Despite increased overall genomic instability and a predominance of copy-number losses over gains in RIG (Fig. 3c), no additional specific focal or large segment alterations were detected in RIG pedRTK I cases relative to pHGG pedRTK I cases (Supplementary Fig. 7 and Supplementary Data 6, 18–20).

To further investigate potential sources of copy-number amplification, the rate of chromothripsis in the RIG samples relative to that in de novo nonbrainstem and brainstem pHGG was analyzed[19]. Chromothripsis increased in the RIG cohort (8/12 cases) relative to that in diffuse intrinsic pontine glioma (DIPG) (66.7% vs. 30.0%, $P = 0.048$), nonbrainstem pHGG (66.7% vs. 33.3%, $P = 0.091$), and all primary pHGG combined (66.7% vs. 31.4%, $P = 0.036$) (Fig. 3d and Supplementary Data 7). We also identified two examples of chromothripsis-derived extrachromosomal circular DNA that led to the amplification of *PDGFRA* and *CDK4* in two RIG samples (Supplementary Fig. 8)[20].

Our results support that copy-number gains and losses occur more frequently in RIG than de novo pHGG tumors. The gains and losses, respectively, frequently involve known oncogenes and tumor suppressors, suggesting the importance of this form of genetic alteration in RIG oncogenesis. Further, the increased frequency of chromothripsis in RIG compared to de novo pHGG for two known oncogenes *(PDGFRA, CDK4)* provides a potential mechanistic explanation for the observed copy-number amplification of these genes.

**RIGs have distinct focal abnormalities from those in pHGG.** Focal genetic alterations in the RIG cohort were analyzed using available WES/WGS data from 18 RIG cases (Fig. 4). Tier 1 mutations in the five non-hypermutator WES cases without matched germline data are shown in Supplementary Data 8; Tier 1 mutations for the nine non-hypermutator WGS cases with matched germline data are shown in Supplementary Data 9. No pathogenic germline alterations were noted in the ten RIG cases with matched germline WGS data other than those associated with the hypermutator case (Supplementary Data 10).

Recurrent somatic alterations and somatic variation frequency in the RIG and HERBY cohorts were compared[18]. Somatic variant frequency, measured by total variants per megabase (Mb),

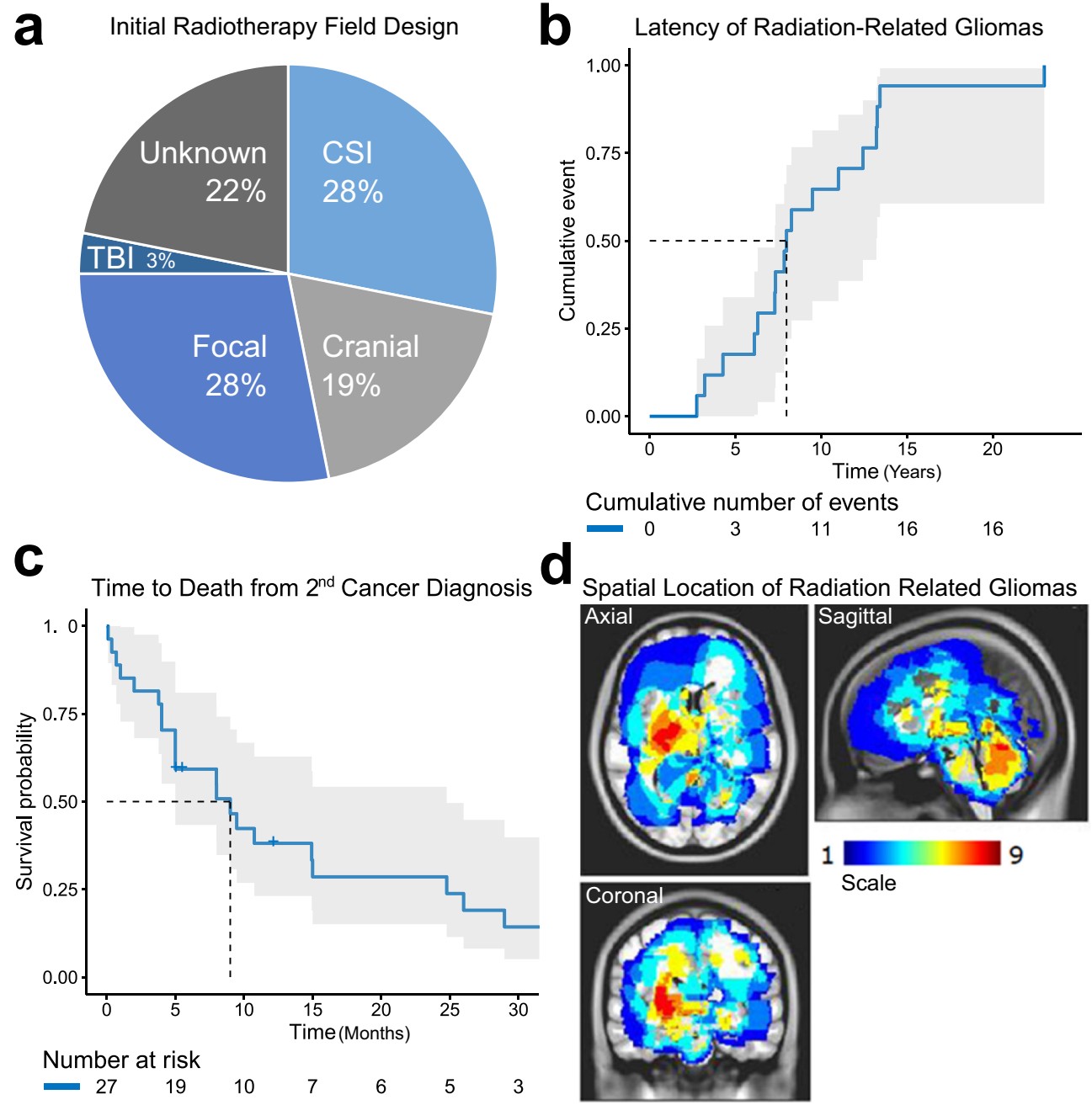

**Fig. 1 RIG cohort characteristics. a** Radiotherapy field type used to treat the initial (pre-RIG) cancer. **b** Time to the diagnosis of RIG from initial cancer diagnosis. **c** Time to death following RIG diagnosis. Two patients survived beyond 35 months but were deceased at study closure. **d** Anatomic location of RIG. Color scale indicates the number of cases anatomically overlapping at each point in space. RIG radiation-induced high-grade glioma, CSI craniospinal irradiation, TBI total body irradiation. The gray region around each line in (**b**, **c**) represents the 95% confidence interval around the estimate.

was significantly greater in the coding regions of RIG DNA compared to pHGG from the HERBY cohort (Supplementary Fig. 10A). Relative frequencies of somatic noncoding base transitions were decreased for A to C and A to G but decreased for C to A transitions in RIG compared to pHGG (Supplementary Fig. 10B, C).

The most frequent recurrent focal somatic alterations in RIG were in *PDGFRA*, *CDKN2A*, *BCOR*, *NF1*, *TP53*, and *CDK4* (Fig. 4 and Supplementary Data 4, 6). Compared to pHGG from the HERBY cohort[18], there were statistically significant increases in

*BCOR* ($P = 0.0004$) alterations. (Supplementary Data 6). The RIG cohort included mutations at several sites that were also commonly observed in pHGG, including *TP53*, *NF1*, and *MET* fusion products (Fig. 4, Supplementary Fig. 9, and Supplementary Data 6, 11, 12). Of note, none of the patients with NF1 somatic mutations met the clinical criteria or had positive germline testing for neurofibromatosis. For other mutations commonly associated with de novo pHGG, only one *H3F3A-K27M* mutation (1/19, 5.6%) was observed in the RIG cohort compared to 37.8% (28/74) in the HERBY cohort of nonbrainstem pHGG ($P = 0.0053$). The

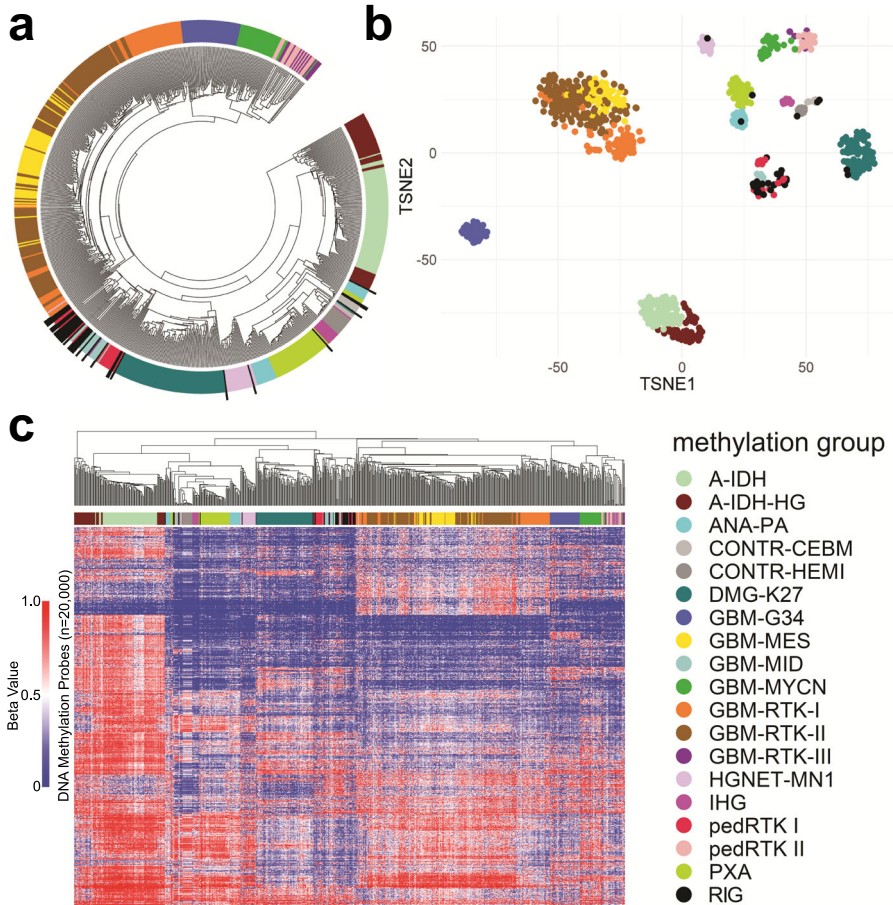

**Fig. 2 Methylation group-based classification of RIG. a** Circular dendrogram indicating the location of RIG (black bars) relative to reference CNS tumors. **b** Localization of RIGs relative to other CNS cancers in t-SNE space (legend shown below). **c** Dendrogram and heatmap of RIG cases clustered against reference CNS tumors (legend to right). A-IDH astrocytoma, subclass IDH-mutant, A-IDH-HG high-grade astrocytoma, subclass IDH-mutant, ANA PA anaplastic pilocytic astrocytoma, CONTR-CEBM control cerebellum, CONTR-HEMI control cerebral cortex, DMG-K27 diffuse midline glioma, H3K27M-mutant, GBM-G34 glioblastoma, subclass H3.3 p.G34R-mutant, GBM-MES glioblastoma, subclass mesenchymal, GBM-MID glioblastoma, IDH-wild type, subclass midline, GBM-MYCN glioblastoma, subclass MYCN-amplified, GBM-RTK-I, adult glioblastoma, subclass RTK I, GBM-RTK-II adult glioblastoma, subclass RTK II, GBM-RTK-III adult glioblastoma, subclass RTK III, HGNET-MN1 high-grade neuroepithelial tumor with MN1 alteration, IHG infantile high-grade glioma, pedRTK I pediatric glioblastoma, subclass RTK I, pedRTK II pediatric glioblastoma, subclass RTK II, PXA pleomorphic xanthoastrocytoma, RIG radiation-induced high-grade glioma, t-SNE t-distributed stochastic neighbor embedding.

RIG cohort did not include *HIST1H3B, IDH1, ACVR1*, or *H3F3A-G34R/V* mutations. In summary, although there were no germline pathogenic mutations in the nine non-hypermutator RIG cases for which we performed germline sequencing, RIG has an increased burden of known oncogenic somatic alterations (Supplementary Figs. 9 and 10A), with many alterations occurring in known oncogenes such as *PDGFRA, BCOR, CDK4, TP53*, and *NF1*.

**RIGs cluster separately from pHGG based on gene expression.** Transcriptomic data (microarray or RNA-seq) were available for 13 RIG cases and 42 de novo glioblastoma (GBM) cases treated at Children's Hospital Colorado. Based on our analysis of gene expression in these cases, RIGs clustered separately from de novo pediatric, infant, and adult GBM and formed two distinct subgroups (A and B) that included six and seven tumor samples, respectively, with broad differences in gene expression (Fig. 5a, b). A comparison of the transcriptomic and methylation-based clustering results showed that the two expression-based RIG subgroups were also reflected in the methylation analysis with 5/6 Group A RIGs and 5/7 Group B RIGs clustering together in the methylation data (*P* < 0.05) (Fig. 5c). Metascape/Cytoscape[21] analysis suggested differences between RIG and the de novo GBM

tumors in basic cellular processes, including RNA processing and transport, protein translation and catabolism, cellular signaling, and pathways controlling neurogenesis, and gliogenesis (Fig. 5d and Supplementary Data 13). To further explore the gene expression patterns identified in Metascape/Cytoscape analysis, geneset enrichment analysis (GSEA) was performed using the gene ontology (GO) geneset collection[22–25]. Compared to de novo GBM, RIGs were enriched in DNA metabolism, cell cycle progression, DNA repair, nervous system development, and protein catabolism, and depleted in immune response, signaling and cellular response to external stimulus, receptor activity, and neurogenesis (Fig. 5e and Supplementary Data 14).

**Expression-based characterization of RIG subgroups.** To investigate the factors underlying the division of RIG tumors into two gene expression-based clusters (Fig. 5a), we studied relative gene expression, mutational status, and copy-number alterations by subgroup. GSEA revealed that gene expression in RIG Group A has stem-like (proneural) and neuronal characteristics and enriched expression of *MYC*-pathway genes (Fig. 6a). In contrast, expression Group B has mesenchymal and astroglial characteristics, enriched expression of inflammatory (particularly NF-κB

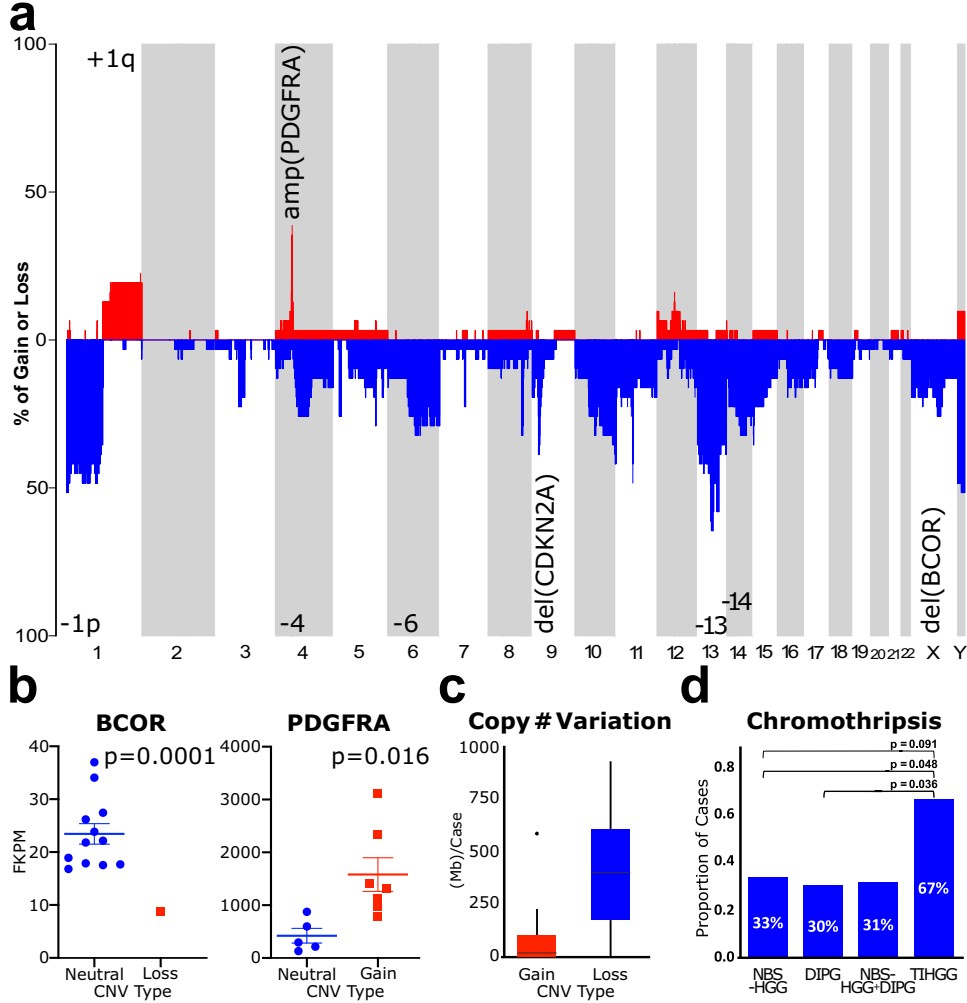

**Fig. 3 Frequent copy-number alterations in RIG. a** Copy-number frequency plot showing the percentage of samples with gain or loss by chromosome. Locations of specific copy-number alterations discussed in the text are labeled. **b** Expression-based confirmation of high-frequency copy-number alterations in *PDGFRA* ($P = 0.016$) and *BCOR* ($P = 0.0001$); mean and SEM are shown. **c** Frequency of copy-number gain and loss across all RIG; boxes show median and first and third quartiles, with whiskers representing the range limited to 1.5× the interquartile range from the box edge. **d** Frequency of chromothripsis in RIG vs. primary NBS-HGG, primary DIPG, and combined NBS-HGG and DIPG (one-sided Fisher's exact test). RIG radiation-induced high-grade glioma, NBS-HGG nonbrainstem high-grade glioma, DIPG diffuse intrinsic pontine glioma, pHGG pediatric high-grade glioma, SEM standard error of the mean. In panel **b**, the *P* value for *BCOR* is computed using Student's single-sample, two-sided *t* test; the *P* value for *PDGFRA* is computed using Student's two-sample, two-sided *t* test.

pathway) and immune genes, and depletion of DNA-repair pathway genes (Fig. 6a and Supplementary Data 15). There were no differences in relative expression between the RIG subgroups in cell cycle or proliferative genes, consistent with clinical experience showing that RIGs, in general, are highly proliferative.

Using DNA methylation data, differences in copy-number alterations between the two gene expression subgroups were analyzed. Group A was enriched in Ch. 1p loss (5/6 tumor samples) compared to Group B (0/7 tumor samples, $P = 0.005$) (Supplementary Data 1). GSEA showed differences at two specific locations in Ch. 1: 1p34 (normalized enrichment score ($NES$) = −5.21, false discovery rate ($FDR$) = 0) and 1p36 ($NES = -7.77$, $FDR = 0$). The remaining gene- or chromosome-level amplifications and deletions identified in the RIG cohort, including Ch.13 or 14 loss, *PDGFRA* amplification, and *CDKN2A* loss, were relatively evenly distributed between the two RIG subgroups (Supplementary Data 1). No other patterns of significant point mutation or similar small-scale genetic differences between

Group A and B tumors were identified and there was no significant relationship between the initial malignancy and RIG subgroup. However, Group A tended to have hematopoietic initial malignancies, whereas Group B tended to have medulloblastomas (Fig. 4 and Supplementary Table 5).

To understand the potential impact of the relative depletion of DNA-repair pathway genes in Group B, gene variant frequencies in the two RIG subgroups were investigated. Except for one hypermutator case, WGS data were available for matched germline (from blood) and tumor samples for three cases from Group A and six from Group B. Despite nearly identical variant frequencies in Group A and B germline samples, Group B tumors had a ninefold greater somatic variant frequency than did Group A tumors ($P < 0.002$) (Fig. 6b).

As noted previously, Group A tumors were enriched in DNA-repair pathway gene expression compared to Group B (mean $NES = -4.55$, $P = 0.0001$ vs. expected value of 0). Likewise, germline (blood) samples from Group A patients were enriched

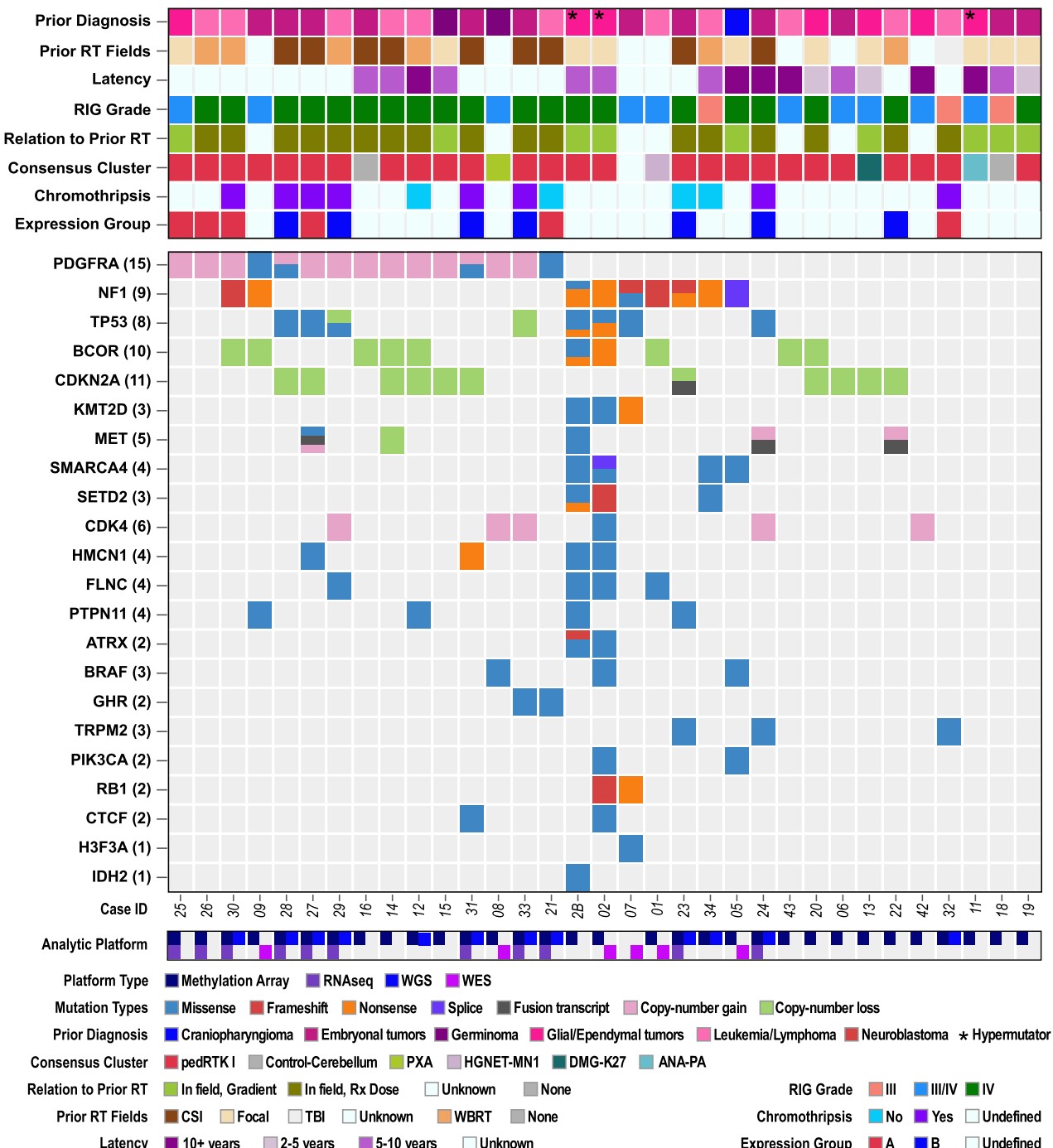

**Fig. 4 Recurrent molecular alterations in RIG.** Summary by RIG sample of clinical characteristics, histopathological features, methylation profile, tier1 mutations, genes affected by copy-number gain/loss, and fusion genes. The second sample for Case 2 is shown as Case 2B, whereas the second samples for Case 14 and Case 12 are not shown but are annotated in Supplementary Data 1. ANA PA anaplastic pilocytic astrocytoma, CONTR-CEBM control cerebellum, DMG-K27 diffuse midline glioma H3.3 K27M, GBM-MID glioblastoma IDH-wild type, subclass midline, HGNET-MN1 high-grade neuroepithelial tumor with MN1 alteration, PXA pleomorphic xanthoastrocytoma, RIG radiation-induced high-grade glioma. Sequencing/array platforms for each case are also shown in the bottom row.

in DNA-repair pathway gene expression (Fig. 6c). Using DNA-repair genesets available in the MSigDB (Broad Institute), individual genes appearing in multiple genesets were identified, and their expression correlated with DNA repair (Fig. 6d)[24,26–31]. In a pairwise comparison, mean fold change (Group B/Group A) of genes shown in Fig. 6d was 0.76 ($P = 9.79 \times 10^{-11}$ vs. expected value of 1) (Supplementary Table 4).

In summary, RIGs split into two transcriptional subgroups with distinct gene expression profiles compared to de novo GBM. Subgroup A resembles proneural GBM and is enriched in the expression of *MYC*-pathway genes, whereas subgroup B resembles mesenchymal GBM and has the greater mutational burden and decreased DNA-repair gene expression.

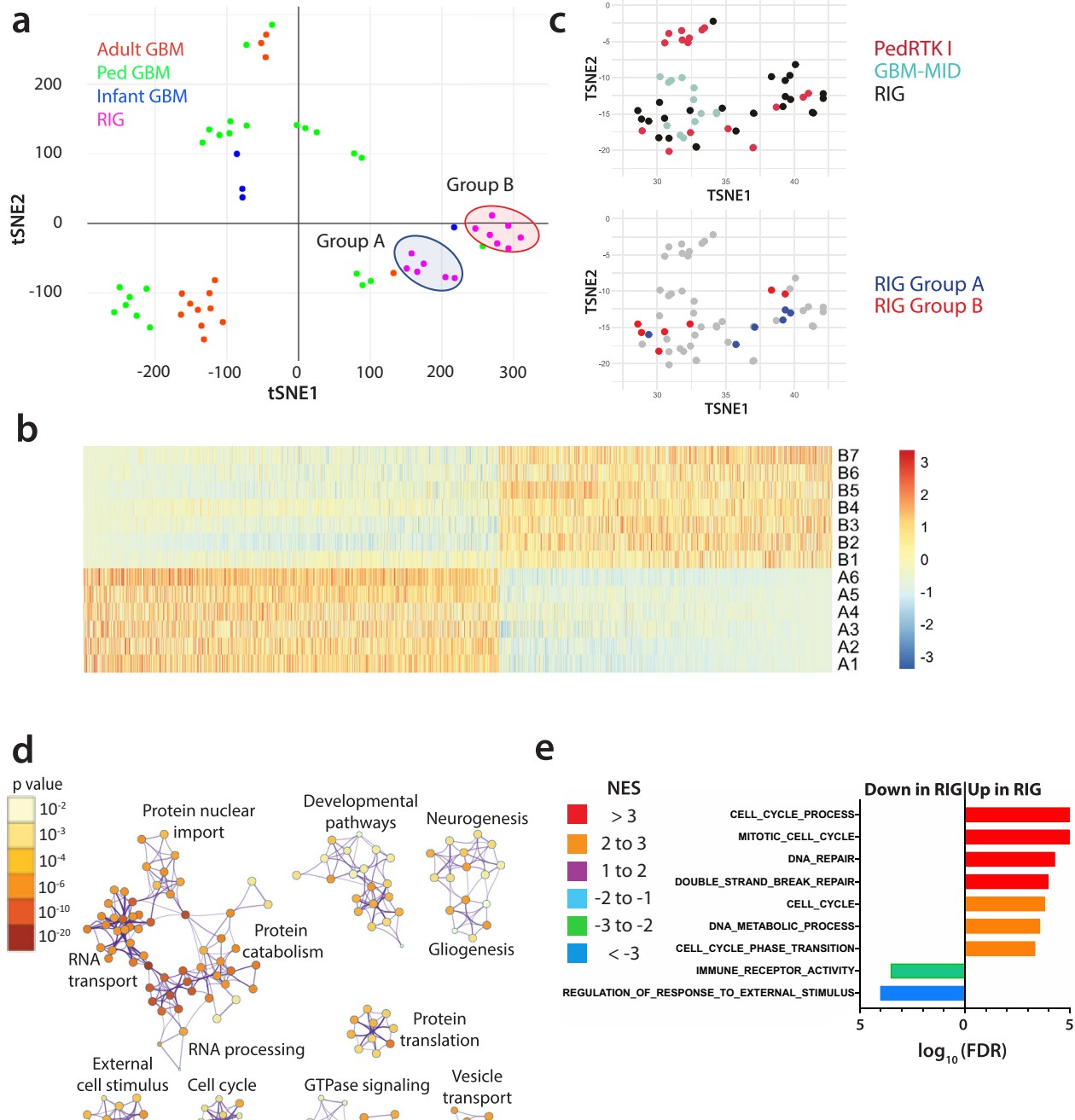

**Fig. 5 Gene expression profiling of RIG versus de novo GBM. a** Clustering of RIGs with microarray-based transcriptomic data ($n = 13$) versus de novo pediatric GBM ($n = 24$), infant GBM ($n = 4$), and adult GBM ($n = 14$) using t-SNE analysis. **b** Heatmap of genes whose mean expression differs between RIG Groups A and B ($n = 2961$, $P < 0.05$) (Labels A1–A6 represent subgroup A RIG tumors and B1–B7 represent subgroup B tumors). The scale represents fold change of Group A vs. Group B. **c** Upper panel: methylation clustering showing locations of RIGs versus PedRTK I and GBM reference clusters; Lower panel: methylation-based clustering of RIGs showing transcriptomic group A and group B locations. Despite overlap of a few samples, group A and group B clustered separately by methylation ($P < 0.05$); **d** Metascape analysis based on GO genesets shows cellular pathways and processes that differ based on gene expression between RIG and de novo GBM (from panel A); color scale represents the $P$ value; comparisons are non-directional between sample sets. **e** GSEA using GO genesets identifies differences in gene expression between RIG and de novo GBM (from panel A); number scale represents the log of the FDR (with FDR = 0 set to a value of 5 for purposes of plotting); colors represent the ratio of GSEA normalized enrichment score (NES) of RIG/de novo GBM. GO gene ontology, GSEA geneset enrichment analysis, RIG radiation-induced high-grade glioma, GBM glioblastoma, FDR false discovery rate.

**In silico and in vitro drug screening**. To identify potential therapeutic susceptibilities in RIG, Metascape and GSEA were used to identify upregulated gene expression pathways in RIG versus normal pediatric cortical tissue (Fig. 7a and Supplementary

Fig. 11A). Because most anticancer drugs inhibit their targets, the focus was on potential oncogenes and oncogenic pathways, that is, genes with upregulated expression in the Metascape analysis and genesets with positive normalized enrichment scores (NES)

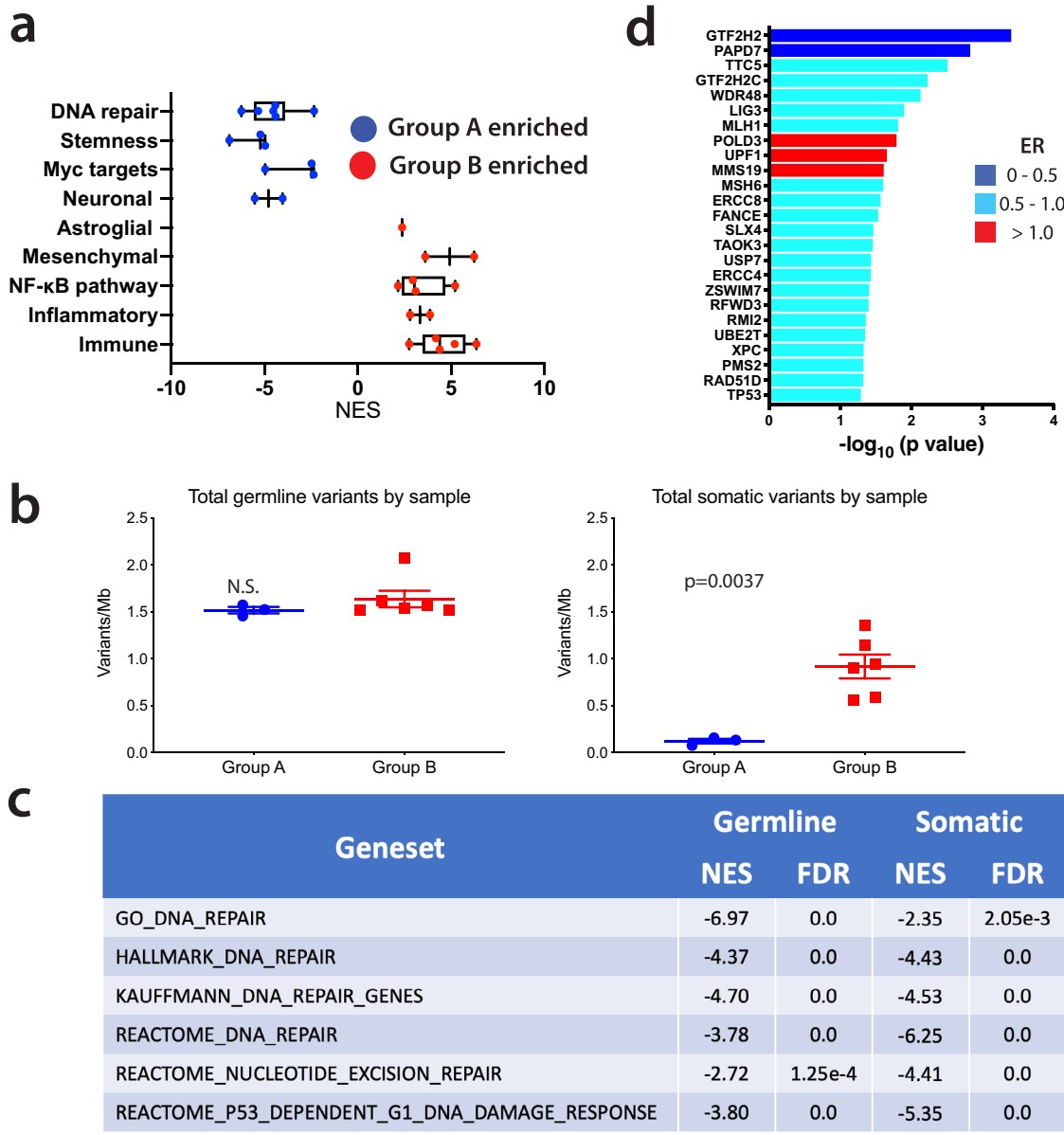

**Fig. 6 Gene expression and genetic profiling of Group A versus Group B RIG. a** GSEA results showing GO *NES* differences by category between Group A (blue) and Group B (red) RIG tumors; horizontal axis is *NES* for comparing RIG Group B vs. RIG Group A (NES > 0 means enrichment in Group B and NES < 0 means enrichment in Group A; *FDR* values for comparisons are listed in Supplementary Data 15); Box-plot midline shows median; hinges are at 25th and 75th percentiles; whiskers show minimum and maximum *NES* values for each geneset. **b** For RIG samples with germline and somatic genome sequencing and transcriptomic data, germline variant load was identical between subgroups, but the somatic load was approximately ninefold greater in Group B (P = 0.0037); the RIG sample with *MSH2* mismatch repair defect was excluded from this analysis. **c** GSEA results for DNA-repair pathways for Group B versus Group A germline and tumor samples; negative numbers represent depletion in Group B versus Group A. **d** Differences in expression for pre-selected individual genes from the DNA-repair genesets that were judged to be most reflective of DNA-repair efficiency are shown by decreasing *P* value (top to bottom); colors represent the log$_{10}$ ratio of fold change in Group B versus Group A. NES normalized enrichment score, RIG radiation-induced high-grade glioma, GSEA geneset enrichment analysis, FDR false discovery rate. Significance testing in panels **b** and **d** was performed using Student's two-sample *t*-test; tests were two-sided. Adjustments for multiple comparisons were not performed in part **d** because genes were chosen a priori. *NES* and *FDR* q values were calculated within the GSEA software (see "Methods" for parameters).

in GSEA. Upregulation likely related solely to the fact that tumors have increased proliferation was disregarded. We instead focused on potentially targetable pathways (based on US Food and Drug Administration [FDA] approved anticancer agents). Modeling identified several targetable, potentially oncogenic pathways in RIG versus normal cortical tissue, including DNA damage surveillance/repair, proteasomal activity, Aurora B kinase, and MAPK signaling (Fig. 7a). DNA repair (*NES* = 2.11, *FDR* = 0.047) and oncogenic MAPK signaling (*NES* = 1.57, *FDR* = 0.21)

pathways were also upregulated in RIG vs. de novo GBM (Supplementary Data 16).

An in vitro drug screen was performed using FDA-approved anticancer agents, including several from each of the classes identified in the gene expression analyses (Supplementary Data 17). Results combined from two cell lines (one Group A and one Group B) showed that at least half of the DNA intercalators, microtubule agents, proteasome inhibitors, HDAC inhibitors, and RAF-MEK pathway inhibitors led to 50% or

**a**

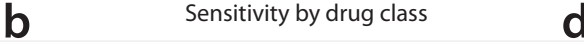

| Geneset | NES | FDR | Size | Drug category |
|---|---|---|---|---|
| GO_DNA_REPAIR | 7.01 | 0 | 480 | DNA Intercalator/ damaging agent |
| REACTOME_HOMOLOGY_DIRECTED_REPAIR | 5.02 | 0 | 98 | |
| PID_BARD1_PATHWAY | 3.64 | 0 | 28 | |
| KEGG_PROTEASOME | 3.53 | 0 | 43 | Proteasome inhibitor |
| PID_AURORA_B_PATHWAY | 3.26 | 7.0E-6 | 34 | Aurora B Kinase inhibitor |
| REACTOME_MAPK_FAMILY_SIGNALING_CASCADES | 1.72 | 0.07 | 85 | MAPK inhibitor |

**b** Sensitivity by drug class

**d**

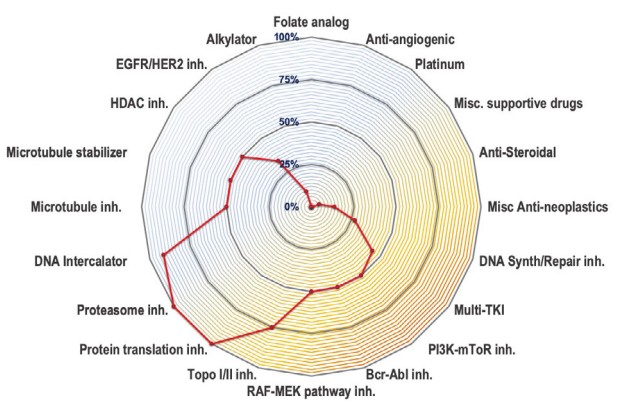

| Drug | IC$_{50}$ | |
|---|---|---|
| | Group A (95% CI) | Group B (95% CI) |
| Aldoxorubicin | 197 (154-251) | 182 (112-298) |
| Bortezomib | 5.38 (4.31-6.92) | 10.3 (10.0-10.7) |
| Carfilzomib | 0.893 (0.693-1.28) | 4.06 (3.79-4.42) |
| Etoposide | 286 (236-345) | 797 (367-1780) |
| Marizomib | 366 (182-1136) | 379 (260-586) |
| Paclitaxel | 82.9 (51.1-132) | 26.9(14.3-50.6) |
| Trametinib | 57.5 (32.4-106) | 93.9 (71.2-121) |
| Vinblastine | 3.28 (2.60-4.16) | 1.05 (0.73-1.61) |

**c** *In vitro* drug screen

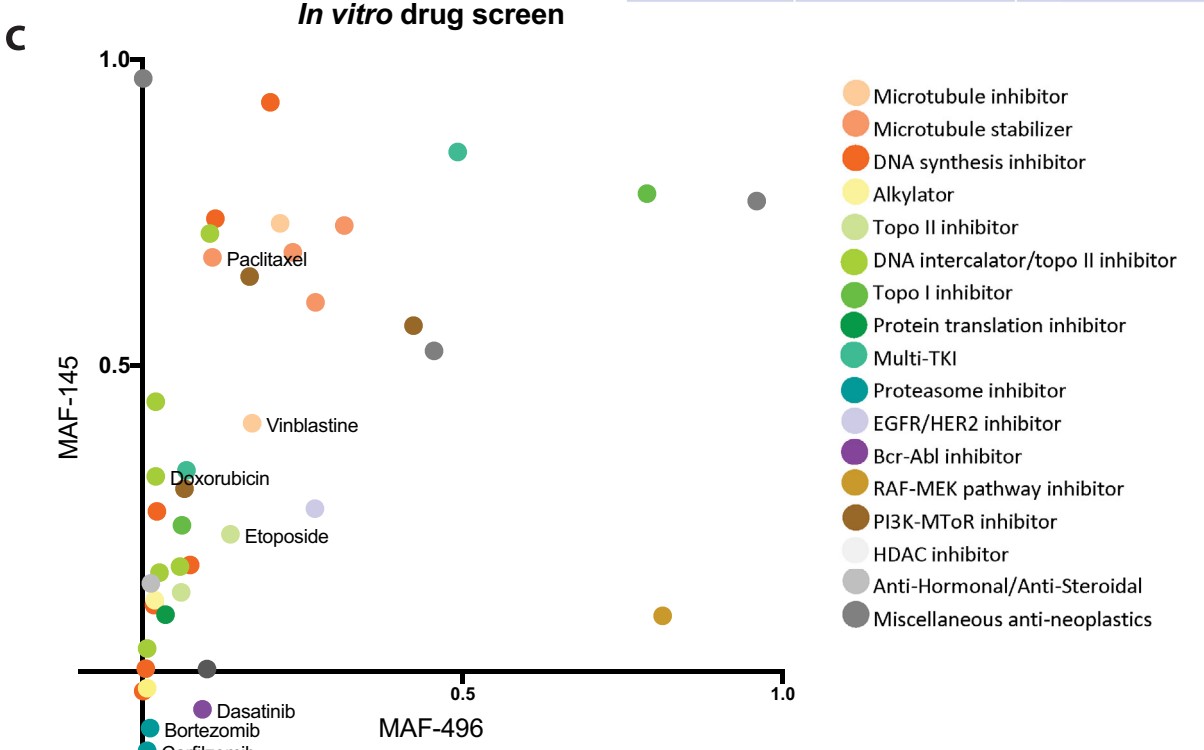

greater cell death compared to the vehicle (Fig. 7b and Supplementary Data 17). Folate analogs, platinum-based drugs, and alkylators performed poorly (Fig. 7b and Supplementary Data 17). Except for proteasome inhibitors (further discussed below), the Group A cell line (MAF-496) was generally more susceptible to the effective drugs than the Group B cell line (MAF-145) (Fig. 7c).

To validate the screening results, in vitro assays of selected agents were performed at a range of concentrations. Aldoxorubicin (an anthracycline that can penetrate the blood–brain

**Fig. 7 RIG preclinical drug screen. a** In silico-predicted response of RIG vs. de novo GBM tumors to drug classes based on GSEA (*NES, FDR*, and number of genes in the geneset). **b** In vitro drug-screening results by drug class combined for RIG cell lines MAF-145 (gene expression Group B) and MAF-496 (gene expression Group A) showing percentage reduction in survival by drug class relative to the vehicle; the screen was performed using FDA-approved anticancer agents at a concentration of 1 µM for 120 h. **c** In vitro drug screen results in RIG cell lines using a 1 µM concentration; Group A (MAF-496) cell line response (surviving fraction vs. vehicle) is plotted on the x axis and Group B cell line (MAF-145) on the y axis. Dot colors correspond to drug class as shown in the legend. **d** In vitro validation results in RIG cell lines MAF-145 and MAF-496 for candidate drugs identified through the in vitro drug screen, $IC_{50}$ in nM. GSEA geneset enrichment analysis, NES normalized enrichment score, RIG radiation-induced high-grade glioma, GBM glioblastoma, FDR false discovery rate, *FDA* Food and Drug Administration.

barrier), etoposide, paclitaxel, and vinblastine showed sub-micromolar $IC_{50}$ values in both cell lines (Fig. 7d and Supplementary Fig. 11B, C), as did the MEK inhibitor trametinib (Fig. 7d and Supplementary Fig. 11D). Sunitinib had an $IC_{50}$ of 5256 nM in the MAF-145 (Group B) cell line but did not reach an $IC_{50}$ level in MAF-496 cells (Group A) (Supplementary Fig. 11E). Validation testing of bortezomib and carfilzomib showed in vitro effectiveness in Group B cells (Fig. 7d and Supplementary Fig. 11F). In the Group A cell line, however, 25–45% of cells survived the highest concentration of each drug, suggesting the presence of a substantial drug-resistant population (Supplementary Fig. 11F). The proteasome inhibitor marizomib, which can penetrate the blood–brain barrier, had $IC_{50}$ values of 366 and 379 nM in Group A and Group B lines, respectively, with a drug-resistant population of ~30% of cells in both cell lines (Supplementary Fig. 11G), which was not seen with bortezomib or carfilzomib. Further investigation showed that bortezomib treatment reduced nuclear levels of NF-κB, a known target of proteasome inhibition, in the Group B cell line (MAF-145), but had a lesser effect on already low nuclear levels of NF-κB in the Group A line (MAF-496) (Supplementary Fig. 11H, I).

Taken together, our drug-screening results in RIG tumors show that drugs that interfere with the S-phase (aldoxorubicin and etoposide) or M-phase (vinblastine and paclitaxel) of the cell cycle and the MEK inhibitor trametinib are effective in vitro in both RIG subtypes. In addition, proteasome inhibition is effective in the Group B cell line.

## Discussion

Our analyses define the molecular characteristics of RIG and its relationship to other forms of pHGG. RIG tumors had defining characteristics independent of their clinical origin. Mutational and transcriptional aspects distinguish RIG from de novo pHGG and provide insights into the origin, clinical course, and treatment ineffectiveness of RIG. Compared to pHGG, RIGs are enriched in DNA repair and cell cycle progression pathways but depleted in immune response and cellular response to external stimuli.

RIGs exhibit recurrent genetic and gene expression alterations. These include loss of Ch.1p, amplification of Ch.1q, *PDGFRA*, and *CDK4*, copy-number losses in tumor suppressors such as *CDKN2A* and *BCOR*, and pathogenic mutations in *TP53, NF1*, and *MET* (fusion events). *BCOR* alterations in RIG are not present in tumors in the pedRTK I pHGG methylation subgroup[32] and are typically accompanied in pHGG by co-segregating *HIST1H3B* K27M mutations. Whereas *BCOR* alterations in pHGG result primarily from frameshift or nonsense mutations and are often coupled with loss of heterozygosity[33], *BCOR* alterations in RIG frequently result from nonfocal chromosomal events leading to *BCOR* loss, suggesting mechanistic differences in the origins of the alterations[32]. In RIG, the co-occurrence with *BCOR* loss of molecular alterations facilitating MAP kinase pathway activation and *CDKN2A* loss suggests a potential means of preventing oncogene-induced senescence[34].

We observed both functional and structural characteristics typical of altered DNA-repair capabilities in RIG. Somatic alterations are increased in group B RIG[9], typically in noncoding regions, which is common in syndromes associated with genomic instability[35], because mutations in coding regions are more likely to produce negative selection pressure. The increased somatic alteration load of group B RIGs could arise through the combination of mutagenic treatment-induced DNA damage from RT and downregulation of DNA-repair pathways. Further insights into potential germline susceptibilities in group B patients related to impaired DNA repair might help identify patients at risk for RIG before tumor therapy and allow treatment modification to prevent RIG. Based on the absence of clear germline predictors of increased RIG susceptibility, we hypothesize that additional undescribed or lower penetrance pathogenic alterations may contribute to the development of RIG only in the context of highly mutagenic treatments such as radiotherapy. In contrast to Group B, Group A RIGs have large-scale chromosome-level abnormalities associated with poor outcome in pediatric brain tumors, including Ch. 1p loss and Ch. 13 loss[36,37]. Specific associations between chromosome-level abnormalities and other characteristics of group A tumors have not been reported and need further study.

Several observed instances of oncogene amplification (*PDGFRA* and *CDK4*) may arise from their inclusion in extra-chromosomal circular DNA (eccDNA), which can facilitate independent and more efficient amplification compared to chromosomal copy-number alterations and has been identified as a mechanism of resistance to targeted therapies[20,38–40]. Large-scale localized DNA damage in the form of chromothripsis can also cause carcinogenesis after radiotherapy, as it can perpetuate a string of subsequent random molecular alterations. We observed an increased rate of chromothripsis in RIG with concurrent *TP53* mutations, compared to that of pHGG (which itself has an increased rate of genomic instability due to irregularities in chromatin-modification pathways)[19]. Although chromothripsis could not be evaluated in non-WGS cases, the high frequency of CNVs in the cases that underwent methylation analysis suggests that genomic instability is a feature of RIG.

Our data and analyses identify several important molecular differences between RIG and de novo pHGG. However, differentiating RIG from recurrent, transformed, or de novo pediatric brain tumors by molecular characteristics remains challenging. The RIGs in our cohort are indistinguishable from the DNA methylation subgroups pedRTK I and IDHwt pHGG. Two cases of RIG (cases 7 and 13) had characteristics of diffuse midline glioma (DMG), but only Case 7 bore the defining *H3K27M* mutation. Others have also reported this finding[41]. Our work identifies epigenetic (pedRTK1 vs. other), expression (Group A vs. B), point mutation (*NF1, BCOR*), structural (chromothripsis, *PDGFR/CDK4* amplification, loss of Chr. 1p, *CDKN2A* loss), and clinical (latency, location, dosimetry) aspects of RIG that can prove useful to distinguish RIG from de novo pHGG and recurrent pediatric brain tumors. Future studies evaluating the mutational signatures of pHGG (primary and recurrent) along with RIG may provide further advances in this area[42].

In silico and in vitro drug screening identified several FDA-approved drugs that merit further study as potential therapeutic agents for RIG. Drug classes identified as effective against RIG in

our screens (DNA-damaging agents, anti-mitotic drugs that target microtubules) have not been clinically effective in de novo pHGG[43]. Mechanistically, Group B RIG may be vulnerable to these DNA-damaging and anti-mitotic agents because of its deficiencies in DNA-repair pathways, as cells with substantial drug-induced DNA damage might be unable to complete mitosis and would thus undergo cell death associated with mitotic catastrophe.

Proteasome inhibitors were effective in vitro in the Group B cell line, possibly through a mechanism involving inhibition of NF-κB-mediated inflammatory pathways. Clinically, proteasome inhibition as a therapeutic target has been established in multiple myeloma and mantle cell lymphoma; however, its effectiveness is limited by acquired resistance[44]. This experience suggests that the efficacy of proteasome inhibition for RIG depends on identifying effective combination therapies. Targeting the MAPK pathway may also be an effective strategy, given the frequent *NF1* mutations in RIG. Notably, the MEK inhibitor trametinib is effective in vitro in both RIG cell lines[45].

The strengths of our study include the size and multi-institutional nature of the cohort, as well as the comprehensive, orthogonal, unbiased assays performed. Recent characterization of large cohorts of de novo HGG facilitate the comparison of RIG to the HGG landscape in a way that was not previously possible. We also characterize two primary patient-derived RIG cell lines, allowing for interrogation of therapeutic vulnerabilities. Study limitations include the heterogeneity of molecular assays performed across cohort samples due to differences in tissue availability. This has been partially ameliorated by performing separate assays and analyses at only one institution per modality for standardizing results. In vivo models of RIG are still in development but generating patient-derived xenograft and genetically engineered mouse models are possible and should be further investigated. Our study demonstrates key similarities and differences between RIG, de novo pHGG, and recurrent primary brain tumors. Thereby, our analysis provides a backbone for future investigations on RIG biology as well as more efficacious treatment regimens that integrate historic aspects of pHGG treatment with targeted therapies directed at specific molecular alterations and susceptibilities typical to RIG.

## Methods

**RIG case review**. Fifty-four cases were reviewed across multiple organizations (Children's Hospital Colorado (CHCO), St. Jude Children's Research Hospital, Childhood Cancer Survivors' Study (CCSS), the University of Hamburg, and the University of Florida) from 1981 to 2015 to determine cohort eligibility. The initial query of the CCSS institutional tumor tissue bank was based on the history of prior radiation, with subsequent development of HGG. Patients from each institution were enrolled on Institutional Review Board-approved protocols for the harvest and study of tissue for research, including the Colorado Multi-Institutional Review Board (COMIRB 95-500) and SJCRH IRB Number: Pro00007403; Mnemonic: XPD17-029; Reference Number: 001628, and consented to have their tumors and germline samples used for research purposes. After clinical review, seven cases were judged to be recurrent primary pHGG, two cases were recurrent primary ependymoma, one case was a recurrent vs. malignant transformation of a juvenile pilocytic astrocytoma, and one case was a recurrent glioneuronal tumor. Tissues were not available for six cases, and tissues were of insufficient quality or quantity in two cases (Supplementary Fig. 12).

We used a modified version of Cahan's criteria to determine the eligibility for radiation-induced tumors[46]. All radiation-induced tumors arose within the initial irradiated field. Although Cahan's criteria specify that RIG must have a histologically proven difference between the initial and subsequent tumors, seven cases had an initial diagnosis of glial origin (one anaplastic astrocytoma (AA), two astrocytomas, three ependymomas, and one ganglioglioma). The two cases with astrocytoma presented as low-grade pilocytic astrocytomas arising in the optic tract. Both cases were treated with radiotherapy after progressive visual loss, after a failed trial of chemotherapy. The resultant RIGs originated in the posterior fossa and occipital lobe, respectively, both within a region of the normal but previously irradiated brain.

We also included tumors arising in two patients with germline predisposition syndromes in which clinical evidence supported the diagnosis of induction by radiation. One patient with known mismatch repair deficiency syndrome and germline *PMS2* mutation developed a primary AA localized to the right frontal lobe at age 14 years and was managed by gross total resection, followed by erlotinib and adjuvant radiation as a part of SJHG04 (NCT00124657). After 9.5 years of controlled disease, a new tumor was found centered in the left occipital lobe. The area was judged to be of sufficient latency to be designated a RIG. Review of clinical history, radiographic imaging, initial radiotherapy plan, and pathology supported that the subsequent GBM was more likely to be treatment-induced, as it arose in the prior radiotherapy field, differed from the initial AA, arose from a region of normal brain parenchyma (except for previous irradiation), and occurred with significant delay, making a late recurrence of AA exceedingly unlikely. The second patient had a mismatch repair deficiency arising from a heterozygous loss of *MSH*. When the patient presented with Philadelphia chromosome-positive ALL at age 9 years, the patient received a bone marrow transplant and total body irradiation (12 Gy). At age 20 years, the patient developed a right frontal anaplastic oligodendroglioma (WHO grade III). The clinical diagnosis of a radiation-induced tumor was based on the occurrence of oligodendroglioma in the radiation field and the likelihood that radiation-induced DNA damage played a significant role in tumor formation. Details of the clinical history of all reviewed and included cases are in Supplementary Fig. 1.

**RIG material available for analyses**. Deidentified tumor tissue specimens from frozen or formalin-fixed paraffin-embedded (FFPE) sections and frozen patient blood samples were processed after institutional review board approval. Whole-genome methylation analysis was completed in 31 cases. RNA was of sufficient quality for RNA-seq analysis in 14 cases (Supplementary Data 1). Whole-genome sequencing was conducted in 12 frozen tumors and matched blood samples (to obtain germline genomic information) and whole-exome sequencing on five additional FFPE tumor samples[10].

**RIG location mapping**. T1 images were registered to the Montreal Neurological Institute template[47] using ANTsR[48] and analyzed with voxel-based lesion-symptom mapping (VLSM)[49] to assess the similarity between statistical maps by calculating the correlation between t-scores, treating lesion voxels as subjects (Supplementary Fig. 2A).

**Patient-level statistical analysis**. Patient- and sample-level statistical analyses were performed using Rstudio Version 1.1.463. Packages used for the presented analyses included "survminer," "ggplot2," "survival," and "networkD3." Continuous data were described using non-parametric measures of central tendency and tested across strata by using the Wilcoxon–Mann–Whitney test. Frequency data across groups were evaluated using the Fisher's exact test or Chi-square test. Time-to-event endpoints were summarized using the Kaplan–Meier estimator. Differences in time to event strata were compared by using the log-rank test.

**Methylation array processing**. Tumor DNA was extracted from FFPE material by using the Maxwell16 FFPE Plus LEV purification kit and the Maxwell16 instrument (Promega, Madison, WI) according to the manufacturer's instructions. Extracted DNA from FFPE tissue underwent quality control assessment by using the Illumina Infinium FFPE QC Assay kit for qPCR. The Delta Cq values for all samples were <4. DNA concentration was assessed using PicoGreen. At least 300 ng of DNA was used per sample for the subsequent bisulfite conversion using the Zymo EZ-96 DNA Methylation kit. Next, the Infinium HD FFPE Restoration kit was used to restore degraded FFPE DNA to a state that is amplifiable by the Infinium HD FFPE methylation whole-genome amplification kit. Restored DNA was then plate-purified (with the Zymo ZR-96 DNA Clean & Concentrator-5), amplified, fragmented, precipitated, re-suspended, and hybridized to an Illumina Infinium Methylation EPIC 850 K BeadChip array for 22 h and 30 min (by using the Illumina Infinium Methylation EPIC assay kit). After hybridization, arrays were manually disassembled and washed. Subsequent X-Staining of array features was processed on a Tecan Freedom Evo robotics system. Arrays were then manually coated and imaged using an Illumina iScan system with an autoloader.

DNA methylation data analysis was performed using the open-source statistical programming language R (R Core Team, 2016). Raw data files generated by the iScan array scanner were read and pre-processed using the minfi Bioconductor package[50]. With the minfi package, the same preprocessing steps as in Illumina's Genomestudio software were performed. In addition, the following filtering criteria were applied: removal of probes targeting the X and Y chromosomes, removal of probes containing nucleotide polymorphism (dbSNP132 Common) within five base pairs of and including the targeted CpG-site, and removal of probes not mapping uniquely to the human reference genome (hg19), allowing for one mismatch. In total, 394,848 common probes of Illumina 450 K and EPIC arrays were kept for clustering analysis.

**Statistical analysis of DNA methylation**. To determine the subgroup affiliation of our RIG samples, the reference DNA methylation cohort published by Capper et al. (GSE90496)[16] and an additional 49 reference pediatric HGGs with known molecular features were used[15]. RIG samples were combined with reference IDATs containing CNS tumors and control brain tissues for unsupervised hierarchical

clustering[16]. The 32,000 most variable methylated CpG probes measured by the standard deviation across combined samples were selected. Pearson correlation was calculated as the distance measured between samples, and unsupervised hierarchical clustering was performed by the average linkage agglomeration method. The probe-level beta values were also analyzed using t-stochastic neighbor embedding (t-SNE)[51]. Hierarchical clustering and t-SNE analyses were repeated by using the top 20,000 most variable methylated CpG probes against a reduced reference set of tumors representing 18 different methylation classes[16]. The reduced set contained normal control methylation classes, high-grade diffuse astrocytic tumors, high-grade neuroepithelial tumors with MN1 alteration, pleomorphic xanthoastrocytoma, and anaplastic pilocytic astrocytoma, pediatric HGG RTK I, and pediatric HGG RTK II. Supervised analysis was performed by using the random forest DNA methylation class prediction algorithm (V11b2) by uploading raw IDAT files to www.molecularneuropathology.org[16].

To compare RIG to de novo pHGG, raw IDAT files generated from Illumina Infinium HumanMethylation450 BeadChip platform were retrieved from the HERBY trial dataset ArrayExpress database (E-MTAB-5552)[18].

**Detection of copy-number alterations with methylation array data.** The CNVs were analyzed with Illumina methylation arrays using the conumee Bioconductor package in R using default settings. DNA copy-number segmentations were retrieved from uploading raw IDAT files to www.molecularneuropathology.org. Segmentation files were then imported to IGV (version 2.4.14) for visualization and identification of CNVs. Focal copy-number alterations were defined as regions that span a small proportion (≤25%) of the chromosome arm, whereas other regions were defined as broad alterations. Mean segment value of −0.2 and 0.2 were used as thresholds for losses and gains, respectively. Copy-number plots were manually examined for selected copy-number alterations. When copy-number information was also available from sequencing data, both results were compared and adjudicated. Adjudicated results are shown in Fig. 4 for selected recurrent somatic alterations. $q$ values for focal, broad copy-number alterations were determined using GISTIC (v2.0.23). GISTIC analysis was performed with the following parameters: 0.9 confidence level, 0.2 amplification and deletion thresholds, 0.25 focal length cutoff, and the gene GISTIC algorithm was flagged. The broad analysis and arm peel advanced parameters were also flagged.

**RNA-seq analysis.** Libraries were prepared by using the TruSeq Library Preparation Kit v2 (Agilent). Directional mRNA sequencing was performed at the University of Colorado Anschutz Medical Campus Genomics and Microarray Core on a HiSeq 2500 sequencing system (Illumina) using single-pass 125 bp reads (1 × 125) and approximately 50 million reads per sample. Resulting data were mapped to the human genome (hg19) by gSNAP, expression (FPKM) was derived by Cufflinks, and differential expression was analyzed with ANOVA in R. Output files contained read-depth data and FPKM expression levels for each sample, and when gene expression levels were compared between groups of samples, the ratio of expression in $\log_2$ format and a $P$ value for each gene was recorded. CICERO was used to detect fusion genes in RNA-seq data[52].

**Analysis of transcriptomic data.** Using microarray data (Affymetrix Human Genome U133 Plus 2.0 Array) previously acquired from tumor samples of patients treated at Children's Hospital Colorado, patterns of gene expression in 13 RIG samples were compared to those of a cohort of non-treatment-induced tumors consisting of 24 primary pHGG, four infant HGG, and 14 adult HGG. Clustering analysis was performed using the t-SNE method available in the RTSNE package and confirmed the RIG subgrouping obtained through t-SNE by using non-negative matrix factorization (NMF)[51,53]. Principal component analysis with 30 initial dimensions preceded the t-SNE analysis, in which a perplexity of 3 and 50,000 iterations were empirically selected as providing optimal results. For the NMF analysis, the identical microarray dataset used in the t-SNE analysis was employed, using $k$ (number of clusters) of 2–5. Metascape analysis was performed on microarray data using as input a list of 2162 genes differentially expressed ($P < 0.01$) between the RIG and HGG samples, followed by Cytoscape to identify differentially enriched pathways[54,55]. GSEA (Broad Institute) was performed using the pre-ranked option to identify the direction of enrichment between the HGG and RIG groups and GSEA and Ingenuity Pathway Analysis (IPA, Qiagen) to identify gene expression patterns within the RIG cohort[24,25]. GSEA results were evaluated using the NES, in which increased expression results in a positive score and reduced expression in a negative score. Scores were considered potentially informative from a statistical perspective if the false discovery rate (FDR) was less than 0.25. Software versions used are as follows: Metascape 3.5; GSEA 4.0.x; MSigDB 7.0 and 7.1; IPA Spring 2017 release.

**Whole-genome sequencing.** WGS library preparation and sequencing were performed by BGI Americas; 100-fold mean coverage data were acquired. BGI performed initial quality testing of the sample, including concentration and sample integrity/purity. Concentration was detected by a fluorometer or microplate reader (e.g., Qubit Fluorometer, Invitrogen). Sample integrity and purity were detected by agarose gel electrophoresis (agarose gel concentration: 1%, voltage: 150 V, electrophoresis time: 40 min). After quality confirmation, 1 μg of genomic DNA was

randomly fragmented by Covaris. The fragmented genomic DNA was selected by the Agencourt AMPure XP-Medium kit to an average size of 200–400 bp. Fragments were end-repaired and then 3′ adenylated. Adaptors were ligated to the ends of these 3′ adenylated fragments to facilitate amplification by PCR. PCR products were purified using the Agencourt AMPure XP-Medium kit. Double-stranded PCR products were heat-denatured and circularized by the splint oligonucleotide sequence. The single-strand circular DNA (ssCir DNA) was formed as the final library qualified via a quality control procedure. Qualified libraries were sequenced by BGISEQ-500. Briefly, each ssCir DNA molecule was formed into a DNA nanoball (DNB) containing more than 300 copies through rolling cycle replication. The DNBs were loaded into the patterned nanoarray using high-density DNA nanochip technology. Finally, pair-end 100-bp reads were obtained by combinatorial Probe-Anchor Synthesis. Following mapping, quality control, and somatic mutation (SNV and INDEL) calling and classification[56,57], tumor and germline reads were mapped to GRCh37-lite and Tier 1 mutations (i.e., coding somatic mutations) were identified. Copy-number alterations (CNA) regions were identified using CONSERTING on paired tumor-germline WGS samples with log ratio >0.25 or < −0.25 reported[58]. Structural variants (SVs) of tumor WGS samples were identified using CRES, based on soft-clipped reads evidence[59]. The SVs that had discordant reads support in the tumor sample but not in the paired germline sample were reported.

**Comparison of mutation frequency in de novo pHGG and RIG by the expression group.** Mutation load, small-scale variants, and SVs in the RIG tumor samples and matched blood samples were analyzed. Based on our initial review of genome-sequencing data, several tumor samples appeared to have anomalously large numbers of mutations. Therefore, a mutation load analysis was performed. vcf files containing QUAL-filtered (threshold of 50) and unfiltered putatively damaging variants were prepared. Germline mutation load was defined as the total number of called variants per sample in the QUAL-filtered dataset. Somatic mutation load was determined from tumor samples without QUAL-filtering. Mutation calls were filtered to require at least four reads per called variant to constitute a mutation. This approach was deemed reasonable because of the likely heterogeneity present in tumor samples, such that a clone constituting a fraction of the total sample could have an allelic-level mutation that would be detected in only a small fraction of the overall reads at a particular locus, resulting in a low QUAL score. Data were tested for sensitivity to determine the thresholds for fold requirement and whether a QUAL threshold should be imposed. The tumor sample from the patient with Lynch syndrome was not included in this analysis, because it had a known DNA-repair defect unrelated to the therapeutic radiation treatment.

**Whole-exome sequencing.** Human genomic libraries were generated using the SureSelectXT kit specific for the Illumina HiSeq instrument (Catalog No. G9611B; Agilent Technologies, Santa Clara, CA), followed by exome enrichment using the SureSelectXT Human All Exon V6 + COSMIC bait set (Catalog No. 5190-9307). The resulting exome-enriched libraries were then sequenced by the Genome Sequencing Facility on a HiSeq 4000 (Illumina). WES mapping and quality assessment have been described previously[56,57]. The tumor reads were mapped to GRCh37-lite, and variants were called by Bambino[23] and annotated by Medal Ceremony as "Gold," "Silver," "Bronze", or "unknown"[60]. "Gold" mutations and variants matching with the COSMIC database were retained[61]. For other coding variants, those that were low-frequency (<0.001) or absent in ExAC/1000Genome/NHLBI databases were reported if the variant was supported with at least five mutant alleles and at least 30% VAF[62,63]. The significance of mutated genes was assessed using the Significantly Mutated Gene test[64]. Mutation frequency and composition were analyzed by comparing the number and type of mutations across primary and RIG samples. The absolute number of mutations and frequencies of base-pair substitutions in SNVs were compared across RIG and primary HGG using a $t$-test and Chi-square test, respectively (Supplementary Fig. 10A and Supplementary Data 8, 9). The frequencies of commonly altered genes in RIG and primary HGG were compared by using Fisher's exact test and are listed in Supplementary Data 6.

**Evaluation of chromothripsis events and structure prediction of eccDNA.** The presence or absence of chromothripsis was evaluated in 12 samples with WGS data (Supplementary Data 7). Four key criteria were used to infer chromothripsis as described by Korbel et al.: oscillating CNA regions, clustering of breakpoints, the randomness of DNA fragment joins, and randomness of DNA fragment order[65]. Chromothripsis was called when at least two criteria were satisfied and further evaluated by manual review. The eccDNA structures were constructed following the procedures described in Xu et al. by identifying the cyclic graphs composed of highly amplified CNA segments and their associated SVs[20].

**FISH analysis.** Dual-color FISH was performed on 4-μm-thick paraffin-embedded tissue sections. Probes were derived from BAC clones (BACPAC Resources, Oakland, CA) and labeled with either AlexaFluor-488 or AlexaFluor-555 fluorochromes. BAC clones were used to construct probes for the following genes: *PDGFRA* (laboratory-developed probe [RP11-231C18 & 601I15]; 4p control (CTD-

2057N12 & CTD-2588A19), and *CDK4* (Empire Genomics, Williamsville, New York, Cat# CDK4-CHR12-20-ORGR). Probes were co-denatured with target cells on a slide moat at 90 °C for 12 min. Slides were incubated overnight at 37 °C on a slide moat and washed in 4 M Urea/2× SSC at 25 °C for 1 min. Nuclei were counterstained with DAPI (200 ng/mL) (Vector Labs) for viewing on an Olympus BX51 fluorescence microscope equipped with a 100-watt mercury lamp; FITC, rhodamine, and DAPI filters; 100× PlanApo (1.40) oil objective; and a Jai CV digital camera. Images were captured and processed using the Cytovision software from Leica Biosystems (Richmond, IL)[66].

### In silico and in vitro drug screening and validations

*In silico screen.* Pathway expression analyses were performed to compare RNA-seq data for 14 RIG samples with a sample of normal human cortex tissue using Metascape and GSEA (Supplementary Table 20). Because most drugs act as inhibitors, we focused on upregulated genes in the Metascape analyses and on genesets having positive NES in GSEA. Metascape and GSEA using microarray data for 12 RIG samples and 37 de novo GBM samples (23 pediatric, 14 adult), again focusing on upregulated genes in the Metascape analyses and on genesets having positive NES scores in GSEA, were also performed.

*In vitro drug screen and validation.* For the drug screen performed in RIG cell lines MAF-145 and MAF-496, the Approved Oncology Drugs Set VI (National Cancer Institute), comprising 129 drugs, supplemented by selinexor (Karyopharm Therapeutics) and AZD2014 (Astra Zeneca) was used. The complete list of drugs included in the screen is given in Supplementary Data 17. Cells were plated at a density of 5000 cells per well in 90 μL medium in a 96-well treated cell culture plate (Corning #3595) and allowed to adhere overnight. Drugs were applied in 10 μL of medium/1% DMSO at a concentration of 10 μM, resulting in a final concentration of 1 μM and 0.1% DMSO. Cells were incubated in the drug for 5 days. DMSO (0.1%) was used as a control. Cell viability was assayed after 5 days of treatment, using incubation with tritiated thymidine and quantification using a scintillation counter. Results were collected as counts/min and converted to survival by using the formula (sample – medium)/(DMSO – medium), where "sample" is the scintillation count for each drug-treated sample, "medium" is the scintillation count for a well containing medium only, and "DMSO" is the scintillation count for a three-well average of cells treated with 0.1% DMSO only (Supplementary Data 17). The drug screen was conducted twice in MAF-145 cells and once in MAF-496 cells due to limitations on cell availability.

Validation tests of single drugs were conducted using drug concentrations ranging from 0.316 nM to 10 μM in half-log$_{10}$ increments. Cells were plated as described above and incubated in a drug for 120 h. Three biological replicates were used for each drug concentration. Results were assessed using CellTiter 96 Aqueous One Solution Cell Proliferation Assay (Promega), according to the manufacturer's instructions. Survival was computed as above. IC$_{50}$ values were calculated using a variable-slope four-parameter non-linear model with maximum survival constrained at 100% (Prism 7, Graphpad).

*Immunofluorescence staining of in vitro samples.* Cells were plated at a density of 20,000 cells per well in BioCoat chamber slides coated with poly-D-lysine or poly-D-lysine and laminin (Corning) and allowed to adhere for ~24–48 h before being subjected to experimental conditions. Cells to be stained were fixed for 20 min in formaldehyde diluted to 3.7% in PBS (Sigma), permeabilized in 0.1% Triton-X in PBS for 10 min, and blocked for 45 min in 4% BSA in PBS supplemented with 0.05% Triton-X. Cells were incubated in primary antibody to the p65 subunit of NF-κB (Cell Signaling, #6956, 1:400) diluted with 4% BSA (in PBS and 0.05% Triton-X) for 1 h at room temperature or overnight at 4 °C. After multiple rinses with PBS, cells were incubated in a secondary fluorophore (AlexaFluor 488) for 1 h, rinsed, and coverslips were then adhered using ProLong Gold antifade reagent with DAPI (Invitrogen). Confocal imaging was performed at ×400 using 405 nM (DAPI) and 488 nM (AlexaFluor 488) lasers on a 3I Marianas imaging system (Intelligent Imaging Innovations). Images were captured using an Evolve 16-bit EMCCD camera (Photometrics).

**Reporting summary**. Further information on research design is available in the Nature Research Reporting Summary linked to this article.

## Data availability

The WGS and RNA-Seq data generated in this study are available in the European Nucleotide Archive (ENA) under accession code PRJEB32299. The DNA methylation data generated in this study are available in the Gene Expression Omnibus (GEO) under accession code GSE175543. The remaining data are available within the Article, Supplementary Information, or Source Data file. Source data are provided with this paper.

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

## Acknowledgements

We thank members of the Zhang and Wu laboratories (St. Jude Children's Research Hospital) and members of the Foreman and Vibhakar laboratories (University of Colorado) for assistance and discussion. We also thank Matthew Lear (St. Jude Children's Research Hospital) for helping acquire and perform preliminary clinical analysis of primary tumor samples, and Scott Olsen, Granger Ridout, and Emily Walker (Hartwell Center for Bioinformatics and Biotechnology) for their assistance in sequencing samples. RNA-seq library preparation and high-throughput sequencing were conducted at the University of Colorado Cancer Center Genomics Shared Resource, and tumor sample preparation for imaging was performed by the University of Colorado Cancer Center Histology Shared Resource (both funded by the Cancer Center support grant P30CA046934). Microscopy was performed at the University of Colorado Denver Advanced Light Microscopy Core Facility, supported in part by Rocky Mountain Neurological Disorders Core Grant Number P30 NS048154 and by Diabetes Research Center Grant Number P30 DK116073. This work was supported by the National Cancer Institute (CA55727, G.T. Armstrong, Principal Investigator), the Fördergemeinschaft Kinderkrebs-Zentrum Hamburg (US), St. Jude Children's Research Hospital Cancer Center Support (CORE) grant (CA21765, C. Roberts, Principal Investigator), ALSAC, the Morgan Adams Foundation (ALG), and a St. Baldrick's Foundation Scholarship (ALG).

## Author contributions

A.L.G., J.D., and J.T.L. conceived the project and overall experimental design. A.D. assisted with sample processing and data analysis. J.T.L. and C.H. designed and analyzed de novo pHGG and RIG imaging data. A.L.G. and J.T.L. compiled and analyzed RIG clinical and imaging data; L.H. and A.L. analyzed radiation therapy data for the Colorado cohort. M.H. and T.C.H. procured tumor samples. U.S. provided methylation profiling data and clinical information for tumor samples. S.A. processed samples and performed fluorescence in situ hybridization. T.L., Q.T., and B.A.O. analyzed 450/850 K data from RIG and pHGG cases. J.T.L., K.X., G.W., and B.A.O. conceived relevant comparisons to de novo pHGG copy-number and sequencing data. J.D., B.S., K.J., and A.L.G. designed relevant comparisons and analyzed germline and somatic sequencing data, RNA-seq data, and drug-screening results. S.J.B. provided DNA methylation and sequencing data for PCGP cases with a history of therapeutic ionizing radiation. J.D. performed all primary cell line experiments and was assisted by P.F. and R.L. L.M.H., K.D., J.M.L., N.F., R.V., and S.V. assisted with project planning and data analysis. G.W., K.X., K.J., and D.H. performed all computational analyses and interpretation of germline and somatic sequencing data. J.T.L., J.D., and K.X. performed statistical analyses. M.A., G.A., B.A.O., N.F., and S.B. contributed to primary tumor samples and clinical data. K.X. completed the reconstruction of episomes from WGS data. J.T.L., A.G., B.A.O., T.L., J.D., K.X., Q.T., and G.W. helped interpret the experiments. J.T.L. and J.D. wrote the manuscript with input from all coauthors; all coauthors contributed to manuscript revision, which was led by J.T.L., J.D., B.A.O., and A.L.G.

## Competing interests

The authors declare no competing interests.
