## [Peer Review File · Nature Communications]

Reviewers' Comments:

Reviewer #1:

Remarks to the Author:

Despite its interest, the paper remains difficult to read: this is in part due to missing information (numbering of suppl fig, legend poorly informative) and recurrent errors in figure legends. The paper is also too long and lacks concision. I appreciate authors responses to the queries but I still believe that the paper would have gained to focus on radiation induced gliomas. I suggest to reshape the manuscript in a more synthetic form, highlighting the most important results, and omitting anecdotic features and/or non statistically significant results: those can be documented in supplementary files. While the main text would gain to be shortened, the authors should bring more details and accuracy to the legend of the figures (see below) particularly suppl figures. In fact part of the main text could be moved to the legends. The discussion is too long.

Specific points:

- I see that somatic alteration of NF1 were found in 9 cases. Did you could rule out (by clinics or germline analysis) an NF1 disease which represents a high risk for glioma? Two patients had an history of optic pathway gliomas (11 and 13) with a RIG arising at distance (occipital and brain stem), in the "low dose region". It is striking that these patients do not have a somatic NF1 mutation.
- The assessment "TIHGG are differentially spatially distributed relative to de novo pHGG" is too obvious and just common sense. It does not justify per se a paragraph, considering that Radiation induced Glioma occurs in the irradiated area, the topography reflects merely the location of the tumor (posterior fossa in particular related to medulloblastomas)
- While the majority of TIHGG have PedRTKI profile with frequent PDGFRa ampl, other have different profiles (line 182-184). I am wondering if these cases are clearly radio or treatment-induced: in order to give the reader a complete view of each individual case I suggest to put a comprehensive table indicating, in addition to what is reported on table 1, the germline status (cMMRd etc..), the molecular characteristics: mutations, CNV, methylation class, expression (RTKI, and others), type A/B, chromotrypsis Y/N.
- please indicate whether the fusion genes (l 254; Sup table 11) are in frame and result in a protein fusion?
- line 258, The title announces "TIHGGs are enriched for somatic and germline abnormalities": there are mild differences in somatic abnormalities, but I don't see the point for the germline? There several errors in the figures legends
Fig 1- A: this is not a Venn diagram, rather a pie plot. There is a mismatch between B) and D)
Fig 6: ABCD, there is a frameshift: the legend reports A,B,C,D,E
Fig 7: C instead of B
Suppl fig 8 the legend mentions fusion genes but these fusions are not reported here (only in the suppl tables: which one are inframe? and potentially oncogenic?)
Suppl fig 9: please edit the legend and explicit: A) "germline,..:" but I don't see the germline variants.. what is RT (radiotherapy?) please make it congruent with the legend TIHGG vs de Novo pHGG ? C) please complete the legend and colour codes.
Suppl fig 10: F) how do the authors interpret the threshold observed with MAF-145 with proteasome inhibitors. This is unusual in an IC50 curve. H) which was the concentration used?

Reviewer #2:

Remarks to the Author:

The study provided a comprehensive characterization of TIHGGs with additional drug screening. The authors have addressed the majority of previous concerns adequately. Overall, the study is well done and and important. There are a few minor points as follows:

1. The majority of cases are from 1990-2000 or earlier, only 5/33 are after 2000. Is the stark incidence difference significant? Does this mean that the better radiotherapy modalities decrease the TIHGG incidence? It would be more informative if the authors could provide the epidemiological trend of incidence and frequency of TIHGG.
2. The cases no. 11 and 13 have optic nerve gliomas as their primary tumors at a very early age. They may have a high probability of being associated with germline NF1 mutation, which may also predispose to astrocytoma. Availability of paired samples would have made the diagnosis of TIHGG more valid.
3. The authors responded to the missing heatmap with providing it in Fig6b, but it is still a tSNE plot, not a heatmap as claimed.

RESPONSES TO REVIEWER COMMENTS

Reviewer #1 (Remarks to the Author):

Despite its interest, the paper remains difficult to read: this is in part due to missing information (numbering of suppl fig, legend poorly informative) and recurrent errors in figure legends. The paper is also too long and lacks concision. I appreciate authors responses to the queries but I still believe that the paper would have gained to focus on radiation induced gliomas. I suggest to reshape the manuscript in a more synthetic form, highlighting the most important results, and omitting anecdotic features and/or non statistically significant results: those can be documented in supplementary files. While the main text would gain to be shortened, the authors should bring more details and accuracy to the legend of the figures (see below) particularly suppl figures. In fact part of the main text could be moved to the legends. The discussion is too long.

Response: We thank the reviewer for these very helpful comments. We agree the manuscript should focus on radiation-induced gliomas. Although we have kept the one chemotherapy-induced case in the cohort (because of its strong similarity to the other cases and the importance of highlighting the potential contribution of chemotherapy, and also because its removal would require us to repeat nearly all the analyses included in the manuscript), we have greatly de-emphasized this single case. We have had the manuscript revised by a scientific writing editor, which we believe has significantly improved the clarity and concision. Also, we have extensively revised the manuscript to omit anecdotal features and non-significant results, as well as improved the accuracy of data in the manuscript and accuracy/utility in the figure legends. We agree that this is a good approach to clearly present our results. We have also substantially reduced the Discussion and made it more focused.

Specific points:

1. I see that somatic alteration of NF1 were found in 9 cases. Did you could rule out (by clinics or germline analysis) an NF1 disease which represents a high risk for glioma? Two patients had an history of optic pathway gliomas (11 and 13) with a RIG arising at distance (occipital and brain stem), in the "low dose region". It is striking that these patients do not have a somatic NF1 mutation.

Response: We reviewed the nine instances of TIHGGs with NF1 alterations and found either no mention of neurofibromatosis in clinical records or that neurofibromatosis was clinically considered and ruled out. We have now mentioned this point in the manuscript (l.171-172). It is interesting that TIHGGs in the two patients who had initial optic pathway gliomas did not have NF1 mutations. This observation is consistent with our view that TIHGG represents a separately initiated event that does not share oncogenesis with the initial tumor.

2. The assessment "TIHGG are differentially spatially distributed relative to de novo pHG" is too obvious and just common sense. It does not justify per se a paragraph, considering that Radiation induced Glioma occurs in the irradiated area, the topography reflects merely the location of the tumor (posterior fossa in particular related to medulloblastomas)

Response: We agree with the reviewer and have removed this comparison from the manuscript. We find that the overall identification of anatomical locations of TIHGG is useful and have therefore retained that information.

3. While the majority of TIHGG have PedRTKI profile with frequent PDGFRa ampl, other have different profiles (line 182-184). I am wondering if these cases are clearly radio or treatment-induced: in order to give the reader a complete view of each individual case I suggest to put a comprehensive table indicating, in addition to what is reported on table 1, the germline status (cMMRd etc.), the molecular characteristics: mutations, CNV, methylation class, expression (RTKI, and others), type A/B, chromotrypsis Y/N.

Response: We agree that a supplementary table containing this information would be useful for readers. Supplementary Table 5 already includes much of the requested data. We have now added additional items to Sup. Table 5 and summarized its contents in the manuscript for easy access to data.

4. please indicate whether the fusion genes (l 254; Sup table 11) are in frame and result in a protein fusion?

Response: We have added additional information to Supplemental Table 15 to indicate whether the individual fusions are out of frame or in frame and likely to produce a protein product. The means of generating the fusion gene is indicated in column K (DEL (deletion), INS (insertion), INV (inversion), ITX (intra-chromosomal translocation), CTX (inter-chromosomal translocation), and UN (unknown)). Each gene partner (geneA and geneB) is listed according to whether the product results in a coding product (featureA and featureB).

Definitively determining whether a protein product is produced for the specific fusions is infeasible given the material largely came from FFPE cases with limited material.

5. Line 258, The title states that "TIHGGs are enriched for somatic and germline abnormalities." There are mild differences in somatic abnormalities, but I don't see the point for the germline?

Response: We thank the reviewer for pointing this out. We agree with this observation and have removed the reference to germline abnormalities. This was a drafting oversight that we did not catch despite several reviews of the manuscript.

There several errors in the figures legends

Fig 1- A: this is not a Venn diagram, rather a pie plot. There is a mismatch between B) and D)

Fig 6: ABCD, there is a frameshift: the legend reports A,B,C,D,E

Fig 7: C instead of B

Response: Thank you for pointing out these errors, which we have now fixed.

6. Suppl fig 8 the legend mentions fusion genes but these fusions are not reported here (only in the suppl tables: which one are inframe? and potentially oncogenic?)

Response: We have removed the reference to fusions from the legend for Supplementary Fig. 9 (formerly Supplementary Fig. 8). We have added additional information with regard to reading frame of fusions to Supplemental Table 15. Definitively determining whether a protein product is produced for the specific fusions is infeasible given the material largely came from FFPE cases with limited material. The means of generating the fusion gene is indicated in column K (DEL (deletion), INS (insertion), INV (inversion), ITX (intra-chromosomal translocation), CTX (inter-chromosomal translocation), and UN (unknown)). Each gene partner (geneA and geneB) is listed according to whether the product results in a coding product (featureA and feature). While some of the fusions such as those involving the MET

oncogene are known to participate in oncogenic events in tumors within and outside the brain, the significance of others are less clear. We have included the data here to give a comprehensive report of our genomic analysis.

7. Suppl fig 9: please edit the legend and explicit: A) “germline,...” but I don’t see the germline variants.. what is RT (radiotherapy?) please make it congruent with the legend TIHGG vs de Novo pHGG ? C) please complete the legend and color codes.

Response: Supplementary Figure 9 has been renumbered Supplementary Figure 10 due to a change in an earlier supplementary figure. We have clarified and added the requested identification of the descriptors to the figure caption for Supplementary Figure 10A (l. 788) and added a legend for subpart C as requested.

8. Suppl fig 10: F) how do the authors interpret the threshold observed with MAF-145 with proteasome inhibitors. This is unusual in an IC50 curve. H) which was the concentration used?

Response: We thank the reviewer for pointing this out. The observed threshold is actually in MAF-496, the Group A cell line. Our tentative interpretation, which we mention in the manuscript (l. 270-273), is that proteasome inhibitors work through the NF- κ B pathway to inhibit proliferation in TIHGG. In Group B, the mesenchymal subgroup, NF- κ B is an important oncogenic driver. Because proteasome inhibition tends to work well in NF- κ B–driven tumors (by preventing proteasomal destruction of κ B- α , resulting in NF- κ B inhibition), proteasome inhibition is highly effective in MAF-145, the Group B cell line. In the Group A cell line, however, which appears to have a more stemlike phenotype that is not dependent on NF- κ B to proliferate, proteasome inhibition is less effective, resulting in a substantial treatment-resistant population. For data shown in Sup. Fig. 11(H) (renumbered in the current submission), treatment conditions were bortezomib at 100 nM for 16 h. We have noted this in the figure caption.

Reviewer #2 (Remarks to the Author):

The study provided a comprehensive characterization of TIHGGs with additional drug screening. The authors have addressed the majority of previous concerns adequately. Overall, the study is well done and important. There are a few minor points as follows:

1. The majority of cases are from 1990-2000 or earlier, only 5/33 are after 2000. Is the stark incidence difference significant? Does this mean that the better radiotherapy modalities decrease the TIHGG incidence? It would be more informative if the authors could provide the epidemiological trend of incidence and frequency of TIHGG.

Response: This point refers to the initial cancer diagnosis; fewer recent initial diagnoses leading to TIHGGs are likely due to the latency between the initial cancer treatment and TIHGG, which is approximately 7-13 years. Therefore, many patients treated for initial cancers diagnosed since 2000 have not yet had sufficient time to develop TIHGGs. For the actual TIHGG diagnoses, almost all were diagnosed after 2000 and are fairly evenly distributed over this timeframe (Sup. Figure 1). To probe this further, we used NCI’s Surveillance, Epidemiology, and End Results (SEER) registry to look at the incidence of TIHGG over time and found no notable change in incidence over the timeframe of our study. We also found that due to a default setting, the case timelines for which we did not have exact years listed 2000 as the year of initial cancer diagnosis, making it appear as if many of the initial cancers

were diagnosed this year; we have now corrected these cases so that they show only number of years since initial diagnosis, not actual years (Sup. Figure 1).

2. The cases no. 11 and 13 have optic nerve gliomas as their primary tumors at a very early age. They may have a high probability of being associated with germline NF1 mutation, which may also predispose to astrocytoma. Availability of paired samples would have made the diagnosis of T1HGG more valid.

Response: We re-reviewed the clinical records for these two patients, confirmed that there was no other evidence of neurofibromatosis in either case, and added this observation to the manuscript (l 171-72) .

3. The authors responded to the missing heatmap with providing it in Fig6b, but it is still a tSNE plot, not a heatmap as claimed.

Response: We thank the reviewer for the comment. Due to a mix-up, Figure 6 as resubmitted was not our most current version. We apologize for this error and have now provided the correct figure (Figure 6B) that shows the heatmap.

Reviewers' Comments:

Reviewer #2:

Remarks to the Author:

The authors have satisfactorily responded to all my concerns.

Reviewer #3:

Remarks to the Author:

The manuscript 'Comprehensive molecular characterization of pediatric treatment-induced glioma: A distinct entity despite disparate etiologies with defining molecular characteristics and potential therapeutic targets' by DeSisto et al reports on a cohort of second, post treatment gliomas in pediatric patients. This is an important group of incurable tumors to study. The manuscript is well-written.

My main concerns revolve around the heterogenous group of tumors included, with only a small number of tumors that have been profiled with either whole-exome or whole-genome studies. This diminishes the power of the cohort to find recurrent alterations. I am also concerned regarding the results outlined in the paper that report differences in the post-treatment tumors, given the small cohort of tumors with sequencing. It was unclear to me how the authors identified regions with copy-number variation and whether algorithms that incorporated background levels of alterations at regions across the genome were used, and similarly, which statistical methods were used to identify mutations and copy-number alterations that were more enriched post-treatment vs de novo.

I also agree with Reviewer 1 that the inclusion of one patient treated without radiation adds to this heterogeneity and affects the overall analysis, especially since this tumor was one of an already small group of tumors that had whole-exome sequencing performed.

Finally, while the drug-screen is interesting, it is difficult to interpret the specificity of findings to treatment-induced gliomas.

My specific comments are below:

1. The authors include 2 MMR deficient patients in their cohort, concluding that the site of recurrence and the latency make radiation induced glioma more likely than a recurrent tumor. However, children with germline MMR deficiencies can be faced with multiple distinct tumor occurrences. I am not sure that the authors can be sure that these are truly treatment induced gliomas. It may be useful to see if mutational or SV signatures of chemotherapy or radiation can be found in these tumors. Finally the inclusion of these tumors may appear specific mutations or copy-number alterations to be enriched in the post-treatment tumors since they have high tumor mutational burdens as a result of the MMR. Algorithms such as MuTect that take into background levels of mutations and other factors that can increase the chance of finding mutations within a specific gene (ie gene size, replication timing etc) are needed to call statistically recurrent mutations within a cohort.
2. A large proportion of the cohort were evaluated with only methylation and/or RNA-seq rather than WES/WGS. This significantly reduces the power to determine robust signatures of treatment induced changes in the gliomas.
3. What statistical method did the authors use to determine/identify the regions of copy-number alterations that were different from de novo pHGGs? Were algorithms (such as GISTIC) used to determine if areas of copy-number alterations were in fact recurrently altered in treatment induced gliomas at a frequency that was higher than expected given the background rate of alterations, and thus observed at a higher frequency that if they occur by chance? What statistical method was used to detect consistent differences between the treatment induced gliomas vs de novo cancers?

4. Were the SNVs detected in the gliomas enriched with specific mutational signatures?

5. Drug screen – did you have non-treatment induced gliomas to compare to the results to? Without this, it is hard to know how specific the findings are to treatment induced gliomas vs all gliomas vs even normal cells?

Respects to Reviewers comments

1. I agree with Reviewer 1 re: excluding the chemotherapy case from the analysis. My concern is that the overall number of cases that have sequencing is small, and this tumor is one of them. Its inclusion further reduces the power of the genomic analysis to detect consistent features across the radiation induced gliomas.

Responses to Reviewer #3 Comments

The authors thank Reviewer #3 for reviewing and providing helpful comments on our manuscript. Our responses to the comments of Reviewer #3 are stated below.

Reviewer #3 (Remarks to the Author):

The manuscript 'Comprehensive molecular characterization of pediatric treatment-induced glioma: A distinct entity despite disparate etiologies with defining molecular characteristics and potential therapeutic targets' by DeSisto et al reports on a cohort of second, post treatment gliomas in pediatric patients. This is an important group of incurable tumors to study. The manuscript is well-written.

My main concerns revolve around the heterogenous group of tumors included, with only a small number of tumors that have been profiled with either whole-exome or whole-genome studies. This diminishes the power of the cohort to find recurrent alterations.

We agree that more DNA sequencing data would have been useful, but given the limited number of samples with WES/WGS, we elected to also include samples for which only DNA methylation data were available. We feel that the information these samples contributed compensates for the fact that we were unable to obtain DNA for genomic sequencing from them. The methylation data, enabled us to define characteristic common copy number alterations and the relationship to existing methylation classes of pediatric high grade glioma.

I am also concerned regarding the results outlined in the paper that report differences in the post-treatment tumors, given the small cohort of tumors with sequencing. It was unclear to me how the authors identified regions with copy-number variation and whether algorithms that incorporated background levels of alterations at regions across the genome were used, and similarly, which statistically methods were used to identify mutations and copy-number alterations that were more enriched post-treatment vs de novo.

Copy-number variations (CNVs) were identified from the DNA methylation data, which we had on 31/32 samples in the cohort (following removal of the chemotherapy-induced tumor sample). To define the CNVs in the methylation data, we used the conumee algorithm as further described in lines 463-471 of the revised manuscript. The conumee algorithm combines the intensity values of the methylated and unmethylated channel at each CpG and normalizes them against a set of normal controls with flat genomes (typically of greater than 50 samples) to control for probe and sample bias. Next, neighboring probes re combined into bins using thresholds of minimum probes and minimum size (from: <https://bioconductor.org/packages/release/bioc/vignettes/conumee/inst/doc/conumee.html>). Both steps minimize the effect of random variation and further thresholding is performed such that mean segment values of +/- 0.2 are required to call copy number alterations. Based on the suggestions from reviewers, we utilized the conumee output in a subsequent analysis of significant focal and broad copy number alterations using GISTIC. The calculated focal, broad and arm level copy number alterations in

the RIG and HERBY high grade glioma cohorts were reviewed and the identified focal gene and arm level alterations were then hypothesis testing of the frequency of genetic alterations relative to de novo pediatric high grade glioma was completed using either Chi-square or Fisher's exact tests (FET). These comparisons are shown in Supp. Fig. 7 and Supp. Table 9.

I also agree with Reviewer 1 that the inclusion of one patient treated without radiation adds to this heterogeneity and affects the overall analysis, especially since this tumor was one of an already small group of tumors that had whole-exome sequencing performed.

We have removed this sample from the cohort and revised the manuscript as well as all analyses, figures, tables, and supplementary data in which it was included to reflect the change. Given this change, we have also switched from using the new term "treatment-induced high-grade glioma" to the term "radiation-induced high-grade glioma," since now all patients in the cohort received cranial radiation. We discuss this terminology in the Introduction but feel this will be less of a departure from terminology used in the foundational papers on this topic in the literature.

Finally, while the drug-screen is interesting, it is difficult to interpret the specificity of findings to treatment-induced gliomas.

To identify up- and downregulated pathways, we performed *in silico* analyses comparing gene expression in the radiation-induced (RIG) tumors to normal cortical tissue as well as non-radiation induced tumors. Separately, we performed an *in vitro* drug screen in two RIG cell lines. We compared the results to select several agents for additional validation. These data are presented in Fig. 7, Supp. Fig. 11 and Supp. Tables 20-21. While we agree that more samples could have been useful for this analysis, the methodology is appropriately designed to identify potentially effective treatments that are specific to RIG tumors.

My specific comments are below:

1. The authors include 2 MMR deficient patients in their cohort, concluding that the site of recurrence and the latency make radiation induced glioma more likely than a recurrent tumor. However, children with germline MMR deficiencies can be faced with multiple distinct tumor occurrences. I am not sure that the authors can be sure that these are truly treatment induced gliomas. It may be useful to see if mutational or SV signatures of chemotherapy or radiation can be found in these tumors. Finally the inclusion of these tumors may appear specific mutations or copy-number alterations to be enriched in the post-treatment tumors since they have high tumor mutational burdens as a result of the MMR. Algorithms such as MuTect that take into background levels of mutations and other factors that can increase the chance of finding mutations within a specific gene (ie gene size, replication timing etc) are needed to call statistically recurrent mutations within a cohort.

The reviewer's points are well taken. In considering the inclusion of these tumors in the RIG cohort, we initially relied on clinical diagnoses, whose criteria include that the radiation-induced tumor arise in a

previously irradiated field separate from the original tumor location. Our analysis verified that the tumors from MMR-deficient patients were molecularly similar to the RIG cohort and distinct from *de novo* high grade glioma tumors. For these reasons, we determined that the tumors from MMR-deficient patients were appropriately included in the RIG cohort. We further concluded that inclusion of the tumors from MMR deficient patients in the study adds to the knowledge base regarding treatment limitations in this group.

With regard to the suggestion of performing a study to identify mutational signatures, this is an area of interest for the authors as well. We undertook several analyses to identify potential mutational signatures in the present cohort. The results are not sufficiently definitive so at this time we are seeking more samples to refine this work.

The suggestion to use MuTect or similar algorithms is an excellent one that we will pursue in future analyses with a broader cohort. We have incorporated the use of GISTIC to identify specific focal, broad and chromosome level copy number alterations in the RIG and HERBY cohorts. These results are contained in Supp. Fig. 7 and Supp. Table 9.

2. A large proportion of the cohort were evaluated with only methylation and/or RNA-seq rather than WES/WGS. This significantly reduces the power to determine robust signatures of treatment induced changes in the gliomas.

We agree that having more samples with WES/WGS data would be useful. We note, however, that despite the limited number of samples with WGS/WES data in the current cohort that we were able to identify several novel features of RIG tumors based on the DNA methylation and transcriptomic data. We are continuing to seek additional RIG samples from which WES/WGS data can be obtained to expand our cohort.

3. What statistical method did the authors use to determine/identify the regions of copy-number alterations that were different from *de novo* pHGGs? Were algorithms (such as GISTIC) used to determine if areas of copy-number alterations were in fact recurrently altered in treatment induced gliomas at a frequency that was higher than expected given the background rate of alterations, and thus observed at a higher frequency than if they occur by chance? What statistical method was used to detect consistent differences between the treatment induced gliomas vs *de novo* cancers?

The conumee algorithm was used to define copy number alterations within the cohort and this is described in lines 469-478. We have incorporated the use of GISTIC in the updated analysis and the output has been added to the supplementary tables as requested. We have also replaced supplementary figure 7 to illustrate these updated analyses. Hypothesis testing of the frequency of genetic alterations relative to *de novo* pediatric high grade glioma was completed using Fisher's exact (FET) or Chi-square tests. These comparisons are shown in supplementary table 9.

4. Were the SNVs detected in the gliomas enriched with specific mutational signatures?

Thank you for this suggestion. This was an area of interest for the authors as well and, as noted above, we have performed mutational signature analyses on the current cohort. The present analyses, however, do not enable us to draw conclusions about the correlation between specific SNVs and mutational signatures among samples in our cohort.

5. Drug screen – did you have non-treatment induced gliomas to compare the results to? Without this, it is hard to know how specific the findings are to treatment induced gliomas vs all gliomas vs even normal cells?

In our *in silico* analysis of gene expression data we compared the RIG tumors to a cohort of non-treatment induced high grade glioma samples. While this comparison did not identify significant differences in predicted responses based upon our pathway analyses, we felt that the trends were consistent with the results comparing RIG tumors to normal cortical tissue. As noted above, our methodology is appropriately designed to identify effects specific to RIG tumors. We agree that the power of the analysis was diminished by the number of samples available but feel overall that the analysis produced useful results that we were able to confirm using *in vitro* testing.

Respects to Reviewers comments

1. I agree with Reviewer 1 re: excluding the chemotherapy case from the analysis. My concern is that the overall number of cases that have sequencing is small, and this tumor is one of them. Its inclusion further reduces the power of the genomic analysis to detect consistent features across the radiation induced gliomas.

As noted previously, we have removed this sample from the cohort and our results based upon the reviewers' comments.

Reviewers' Comments:

Reviewer #3:

Remarks to the Author:

Thank you to the Authors for restricting this analysis to radiation-induced gliomas. I think that this makes the analysis and results much cleaner. Congratulations to the Authors for putting together this important manuscript. It would be worth checking the figures. It may have been a function of the merging during PDF creation, but all of the labels for the comut plot for Figure 4 were missing in the PDF I reviewed.

RESPONSE TO REVIEWERS' COMMENTS

Reviewer #3 (Remarks to the Author):

Thank you to the Authors for restricting this analysis to radiation-induced gliomas. I think that this makes the analysis and results much cleaner. Congratulations to the Authors for putting together this important manuscript. It would be worth checking the figures. It may have been a function of the merging during PDF creation, but all of the labels for the comut plot for Figure 4 were missing in the PDF I reviewed.

Response: Thank you for your comments; we agree that removal of the non-radiation induced tumor made the analysis clearer. We noticed that labels for some of the figures went missing in our last submission. We believe this was due to some problems with reformatting. We have carefully checked the final versions of the figures in connection with this resubmission to ensure that all information is present.